# HSF-1 promotes longevity through ubiquilin-1-dependent mitochondrial network remodelling

Annmary Paul Erinjeri [1], Xunyan Wang[1], Rhianna Williams[1], Riccardo Zenezini Chiozzi [2,3], Konstantinos Thalassinos [2,3,4] & Johnathan Labbadia [1] ✉

Increased activity of the heat shock factor, HSF-1, suppresses proteotoxicity and enhances longevity. However, the precise mechanisms by which HSF-1 promotes lifespan are unclear. Using an RNAi screen, we identify ubiquilin-1 (ubql-1) as an essential mediator of lifespan extension in worms overexpressing hsf-1. We find that hsf-1 overexpression leads to transcriptional down-regulation of all components of the CDC-48-UFD-1-NPL-4 complex, which is central to both endoplasmic reticulum and mitochondria associated protein degradation, and that this is complemented by UBQL-1-dependent turnover of NPL-4.1. As a consequence, mitochondrial network dynamics are altered, leading to increased lifespan. Together, our data establish that HSF-1 mediates lifespan extension through mitochondrial network adaptations that occur in response to down-tuning of components associated with organellar protein degradation pathways.

Ageing is a major risk factor for chronic morbidities such as cancers, cardiovascular disorders, and neurodegeneration[1]. These diseases contribute to a rising burden on families, communities, and healthcare across the world[1]. Environmental and genetic factors contribute to ageing by influencing a complex network of pathways and processes that drive cellular dysfunction[1]. These include the loss of protein homeostasis (proteostasis), which is characterised by the appearance and aggregation of misfolded and mislocalised proteins within cells and tissues[2].

Cells possess an array of protein quality control mechanisms collectively referred to as the proteostasis network (PN), which act to preserve proteome integrity[3]. The PN coordinates protein synthesis, folding, disaggregation and degradation and integrates components of the translational machinery, molecular chaperones and co-chaperones and the proteolytic systems - the ubiquitin-proteasome system (UPS), and autophagy-lysosomal system - to ensure cell viability[3].

The cytosolic/nuclear arm of the PN is subject to regulation by heat shock transcription factor 1 (HSF-1), which protects the proteome by driving the expression of heat shock proteins (HSPs) that function as molecular chaperones[3,4]. In line with its function, the knockdown of HSF-1 leads to increased protein aggregation, tissue dysfunction and decreased survival, whereas overexpression of HSF-1 maintains proteome integrity, promotes tissue health, and extends lifespan[5]. While it is apparent that increasing HSF-1 activity is beneficial for longevity, our understanding of the mechanisms that act downstream of HSF-1 to prolong healthy tissue function remains limited.

It is widely believed that HSF-1 regulates ageing by upregulating the expression of HSPs. However, in addition to HSPs, HSF-1 also controls the expression of genes encoding cytoskeletal components, metabolic enzymes, ribosomal subunits, chromatin factors and components of the UPS[6,7]. Moreover, recent work has demonstrated roles for autophagy[8], maintenance of the cytoskeleton and lipid regulation[9,10] in HSF-1-mediated lifespan extension. These

[1]Institute of Healthy Ageing, Department of Genetics, Evolution and Environment, Division of Biosciences, University College London, London, UK. [2]Institute of Structural and Molecular Biology, Division of Biosciences, University College London, London, UK. [3]UCL Mass Spectrometry Science Technology Platform, Division of Biosciences, University College London, London, UK. [4]Institute of Structural and Molecular Biology, Birkbeck College, University of London, London, UK. ✉e-mail: j.labbadia@ucl.ac.uk

observations indicate that HSF-1 regulates longevity through mechanisms beyond HSP-mediated chaperoning of the proteome.

Here, we employ an RNAi screen to identify the HSF-1 target genes that promote increased lifespan in *C. elegans* overexpressing HSF-1 (*hsf-1* OE). We find that the sole worm ubiquilin, ubiquilin-1 (*ubql-1*), is necessary for *hsf-1* OE to increase lifespan. Ubiquilins are multifaceted, conserved shuttle proteins that localise to the cytoplasm and nucleus, where they function as chaperones that aid in the degradation of substrates through the ubiquitin-proteasome system and autophagy[11,12]. This is facilitated by their N-terminal Ubiquitin (Ub)-like (UBL) domain, C-terminal Ub-associated (UBA) domain and STI domains, which enable binding to the proteasome, polyubiquitinated chains and potential clients[11–13]. Despite its central role in protein degradation, we find that ubiquilin-1 does not promote longevity by altering general proteostasis capacity. Instead, ubiquilin-1 increases lifespan upon overexpression of HSF-1 by reducing NPL-4.1 levels and promoting mitochondrial network remodelling.

## Results

### Ubiquilin-1 is required for HSF-1-mediated lifespan extension

Overexpression of HSF-1 (*hsf-1* OE) leads to the extension of lifespan in *C. elegans*[14,15]. To better understand the mechanisms that act downstream of HSF-1 to promote longevity, an RNAi screen was performed to determine which HSF-1 target genes are required for the increased lifespan of *hsf-1* OE worms. Our RNAi screen consisted of 96 *C. elegans* genes shown to be directly regulated by HSF-1 under basal or stress conditions (Supplementary Fig. 1a and Supplementary Data 1)[16]. We identified the gene ubiquilin-1 (*ubql-1*) as the strongest modifier of *hsf-1* OE lifespan without comparable effects on the lifespan of wild-type worms (Fig. 1a).

An analysis of all RNAi clones that reduced the lifespan of *hsf-1* OE worms > 20% but did not alter wild-type lifespan, revealed a network (based on co-expression or physical interaction) formed of two clusters: one containing UBQL-1 and SGT-1, and another containing STI-1, DNJ-12, CCT-2, FKB-6 and ABCF-2 (Fig. 1b and Supplementary Data 1). Conversely, we also identified genes whose knockdown increased *hsf-1* OE lifespan > 20% but had an opposing effect on the lifespan of wild-type worms. These were found to contain a small ubiquilin-associated network (co-expression or physical interaction) containing the ribosome-associated proteins RPS-1 and MBF-1[17], as well as the co-chaperone DAF-41/P23 and the protein disulphide isomerase, PDI-3 (Fig. 1c and Supplementary Data 1).

UBQLN1 has been shown to directly interact with SGTA to mediate protein degradation in human cells[18], while STI1 has homology with UBQLN1 and has roles in protein degradation[12]. Together, these findings suggest that UBQL-1 may act as part of, or in parallel to, a wider network of protein folding and degradation complexes to promote longevity upon *hsf-1* OE.

As the strongest hit from our screen, we decided to focus our attention on *ubql-1*. Consistent with previous reports that the *ubql-1* promoter is bound by HSF-1[16] (Supplementary Fig. 1b), *hsf-1* OE worms exhibited increased *ubql-1* expression in early adulthood (Fig. 1d). Furthermore, an assessment of existing *C. elegans* tissue-specific RNA seq-data[19], revealed that *ubql-1* is expressed in all hermaphrodite tissue/cell types, with the exception of arcade cells and valve cells (Supplementary Fig. 1e).

To verify our RNAi screen, we grew animals on empty vector control (L4440) or *ubql-1*(RNAi) and measured survival in two independent *hsf-1* OE lines[10,15]. Knockdown of *ubql-1* suppressed the increased lifespan of both *hsf-1* OE lines tested (Fig. 1e, Supplementary Fig. 1d and Supplementary Data 2). In addition, we assessed lifespan in *ubql-1(tm1574)* mutants that harbour a 755 bp deletion spanning the whole of exons 1 and 2 and express reduced levels of a truncated *ubql-1* mRNA and protein (Fig. 1d, Supplementary Fig. 1c and f, and Supplementary Data 3). Lifespan assays revealed that *ubql-1(tm1574)* mutants

are shorter lived than wildtype controls and that the presence of the *ubql-1(tm1574)* mutation reduces the lifespan of *hsf-1* OE worms to that of wildtype animals (Fig. 1f and Supplementary Data 2). Together, our data establish *ubql-1* as a mediator of HSF-1-mediated lifespan extension in *C. elegans*. Therefore, we sought to investigate the processes through which *ubql-1* impacts longevity.

### *Ubql-1* is not required for *hsf-1* OE to suppress age-related protein aggregation

To better understand the mechanisms by which *ubql-1* functions to suppress ageing downstream of HSF-1, we first investigated whether *ubql-*1 influences HSF-1 activity. Mutation of *ubql-1* did not alter the expression of canonical HSF-1 target genes basally (*hsp-16.11* and *hsp-70*) or in response to heat shock (*hsp-16.11*, *hsp-70* and *F44E5.4*), in wildtype or *hsf-1* OE worms (Fig. 2a–c and Supplementary Fig. 2a, b). These data suggest that *ubql-1* mutants do not block *hsf-1* OE lifespan extension by reducing HSF-1 activity, although we cannot completely rule out more subtle or nuanced changes to HSF-1.

Next, we asked whether *ubql-1* is necessary for worms to manage heat-induced protein folding stress. As expected, *hsf-1* OE worms survived for longer than wildtype worms following heat stress as young adults (Fig. 2d). Despite being shorter-lived, *ubql-1* mutants were more stress resistant than wild-type worms; however, this was not further increased in *ubql-1* mutants upon overexpression of HSF-1 (Fig. 2d). Given that *ubql-1* mutants do not exhibit hallmarks of increased HSF-1 activity (i.e., expression of *hsp-16.11, hsp-70, or F44E5.4* was not elevated in *ubql-1* mutants) these data suggest that *ubql-1* is necessary for *hsf-1* OE to increase stress resistance in worms.

Many factors and pathways can contribute to stress resistance. Therefore, to more precisely examine the effects of *ubql-1* on proteostasis capacity, we took advantage of well-described polyglutamine::YFP-based (PolyQ::YFP) proteostasis sensors[20–22] expressed exclusively in the intestine (Q44) or body wall muscles (Q35). PolyQ aggregation increased with age in both intestinal and muscle tissues (Fig. 2e–g and Supplementary Fig. 2c–e) of wildtype worms and was strongly suppressed on day 3 and day 5 of adulthood by overexpression of *hsf-1* (Fig. 2e–g and Supplementary Fig. 2c–e). Knockdown of *ubql-1* increased polyQ aggregation on day 5 of adulthood within the muscle, but not the intestine, of both wild-type and *hsf-1* OE worms (Fig. 2e–g and Supplementary Fig. 2c). However, the degree to which *hsf-1* OE suppressed polyQ aggregation in muscle and intestinal tissues compared to wildtype counterparts, was the same in both the control (RNAi) and *ubql-1* (RNAi) treatment groups (Fig. 2f, g and Supplementary Fig. 2d, e). This suggests that *ubql-1* is not necessary for *hsf-1* OE to suppress age-related polyglutamine aggregation.

Given that polyQ is an artificial sensor of proteostasis capacity, we reasoned that loss of *ubql-1* may impede the ability of *hsf-1* OE to promote the degradation of endogenous ubiquitylated substrates with age. Therefore, we assessed the abundance of polyubiquitylated, K48-linked ubiquitylated (proteasome directed) and K63-linked ubiquitylated (lysosome directed) proteins[23], on day 1 and day 5 of adulthood.

We observed that levels of high molecular weight, aggregated polyUb, K48-ubiquitylated and K63-ubiquitylated proteins increased with age in all genotypes tested (Fig. 2h, i and Supplementary Fig. 2f–m). However, the level of soluble and aggregated poly-ubiquitylated or K63-ubiquitylated proteins was not consistently altered on day 1 or day 5 of adulthood in *hsf-1* OE or *ubql-1* mutant animals, compared to wild-type controls (Supplementary Fig. 2f–m).

In contrast, the levels of both soluble and aggregated K48-ubiquitylated proteins were elevated in *ubql-1* mutant worms on day 1 and day 5 of adulthood (Fig. 2h, i and Supplementary Fig. 2h, i). Interestingly, *hsf-1* OE did not consistently alter levels of K48-Ub proteins on day 1 of adulthood but did reduce levels of soluble and aggregated K48-Ub proteins in both wild type and *ubql-1* mutant backgrounds on day 5 of adulthood (Fig. 2h, i and Supplementary

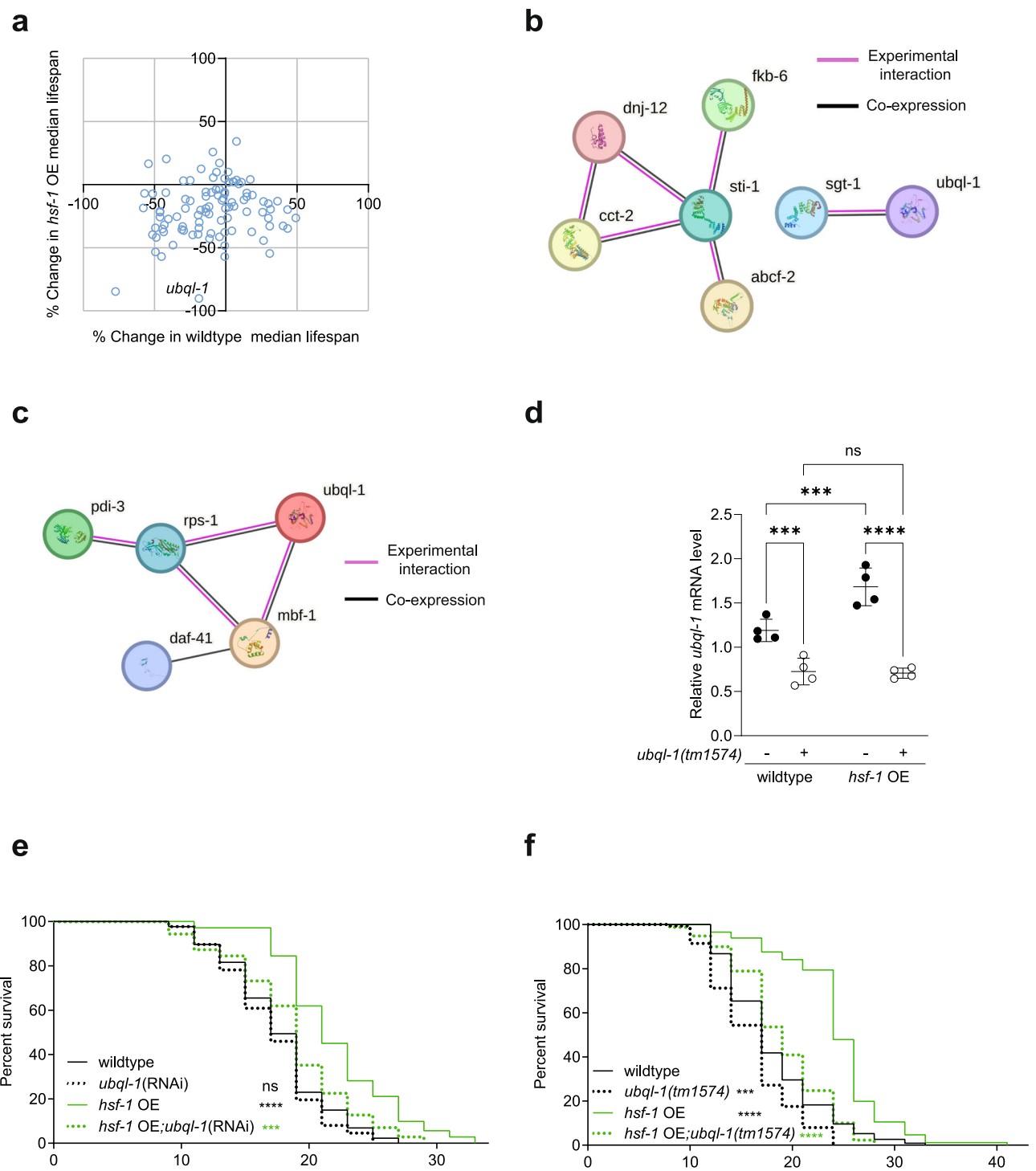

**Fig. 1 | Ubql-1 is necessary for hsf-1 OE to extend lifespan. a** Changes in median lifespan in wildtype and *hsf-1* overexpressing (*hsf-1* OE) worms after RNAi against HSF-1 target genes (sample numbers for all groups can be found within Supplementary Data 1). **b**, **c** STRING networks of screened genes that, when knocked down, were found to (**b**) decrease *hsf-1* OE lifespan by > 20% with no effect on wild-type lifespan, or (**c**) increase *hsf-1* OE lifespan by > 20% but decrease wildtype lifespan by the same degree. Black lines represent interaction by co-expression, and purple lines represent evidence of experimental interaction. *ubql-1* was included in (**c**) for reference, and genes without any connections were removed. **d** Relative expression of *ubql-1* mRNA on day 1 of adulthood in wildtype, *ubql-1(tm1574)*, *hsf-1* OE, and *hsf-1* OE; *ubql-1(tm1574)* animals grown on OP50. Values were normalised to the geometric mean of the housekeeping genes *rpb-2*, *pmp-3,* and *cdc-42*. Data

plotted are the mean of 4 biological replicates +/− SD (wildtype vs *ubql-1(tm1574)*, $p = 0.0008$; *hsf-1*OE vs *hsf-1*OE;*ubql-1(tm1574)*, $p < 0.0001$; wildtype vs *hsf-1*OE, $p = 0.0005$; *ubql-1(tm1574)* vs *hsf-1*OE;*ubql-1(tm1574)*, $p = 0.8662$). **e**, **f** Lifespan of (**e**) wildtype and *hsf-1* OE animals on empty vector and *ubql-1*(RNAi) (wildtype vs *ubql-1*(RNAi), $p = 0.2886$; wildtype vs *hsf-1*OE, $p < 0.0001$; *hsf-1*OE vs *hsf-1*OE;*ubql-1*(RNAi), $p = 0.0002$) and (**f**) wildtype, *ubql-1(tm1574)*, *hsf-1* OE, *hsf-1* OE;*ubql-1(tm1574)* animals grown on OP50 (wildtype vs *ubql-1(tm1574)*, $p = 0.0002$; wildtype vs *hsf-1*OE, $p < 0.0001$; *hsf-1*OE vs *hsf-1*OE;*ubql-1(tm1574)*, $p < 0.0001$). Statistical significance was calculated using (**d**) two-way ANOVA with Fishers LSD and (**e**, **f**) Mantel-Cox log-rank test. ns, not significant ($p > 0.05$), **$p < 0.01$, ***$p < 0.001$, ****$p < 0.0001$. Full statistics for lifespan trials (including n values) can be found in Supplementary Data 2. Source data are provided as a Source Data file.

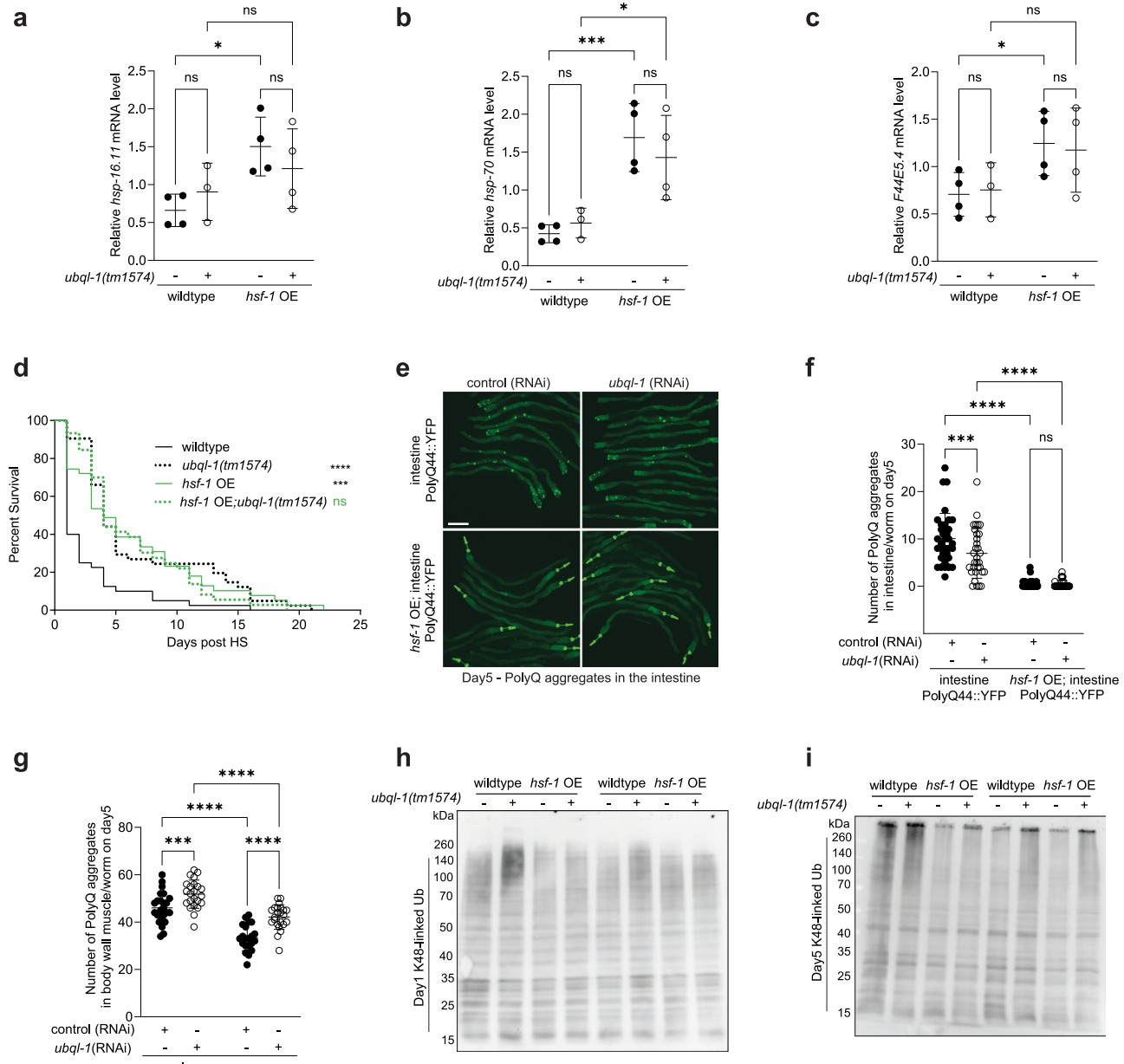

**Fig. 2 | Ubql-1 is not required for hsf-1 OE to suppress age-related protein aggregation. a–c** Relative expression of *hsp-16.11*, *hsp-70* and *F44E5.4* mRNA on day 1 of adulthood in wildtype, *ubql-1(tm1574)*, *hsf-1* OE, and *hsf-1* OE;*ubql-1(tm1574)* animals grown on OP50 and subjected to heat shock (33 °C, 30 mins). Values were normalised to the geometric mean of the housekeeping genes *rpb-2, pmp-3* and *cdc-42,* and data are the mean of 4 biological replicates. Statistical comparisons are: (**a**) *hsp-16.11* (wildtype vs *ubql-1(tm1574)*, $p = 0.4326$; *hsf-1*OE vs *hsf-1*OE;*ubql-1(tm1574)*, $p = 0.3181$; wildtype vs *hsf-1*OE, $p = 0.0115$; *ubql-1(tm1574)* vs *hsf-1*OE;*ubql-1(tm1574)*, $p = 0.3286$), (**b**) *hsp-70* (wildtype vs *ubql-1(tm1574)*, $p = 0.6431$, *hsf-1*OE vs *hsf-1*OE;*ubql-1(tm1574)*, $p = 0.3581$, wildtype vs *hsf-1*OE, $p = 0.0007$; *ubql-1(tm1574)* vs *hsf-1*OE;*ubql-1(tm1574)*, $p = 0.0138$), (**c**) *F44E5.4* (wildtype vs *ubql-1(tm1574)*, $p = 0.8607$; *hsf-1*OE vs *hsf-1*OE;*ubql-1(tm1574)*, $p = 0.7737$; wildtype vs *hsf-1*OE, $p = 0.0463$; *ubql-1(tm1574)* vs *hsf-1*OE;*ubql-1(tm1574)*, $p = 0.1324$). **d** Survival of wildtype, *ubql-1(tm1574)*, *hsf-1* OE, *hsf-1* OE;*ubql-1(tm1574)* animals on D1 of adulthood following exposure to heat shock (35 ˚C for 4 h) (wildtype vs *ubql-1(tm1574)*, $p < 0.0001$; wildtype vs *hsf-1*OE, $p = 0.0001$; *hsf-1*OE vs *hsf-1*OE;*ubql-1(tm1574)*, $p = 0.9380$). **e** Representative images of wildtype and *hsf-1* OE worms expressing intestinal polyQ44::YFP grown on the empty vector (EV) or *ubql-1*(RNAi) at day 5 of adulthood. Scale bar, 200 μm. **f**, **g** Number of polyglutamine::YFP aggregates

present in the (f) intestine (Q44::YFP) and (**g**) body wall muscle (Q35::YFP) on D5 of adulthood in wildtype and *hsf-1* OE animals grown on empty vector and *ubql-1*(RNAi). One of three independent experiments has been represented for intestinal and muscle PolyQ sensors. Intestinal and muscle data are presented as the mean values +/− SD. Statistical comparisons are (**f**) iPolyQ EV (*n* = 35) vs iPolyQ;*ubql-*1(RNAi) (*n* = 32), $p = 0.0009$; *hsf-1*OE;iPolyQ EV (*n* = 33) vs *hsf-1*OE;iPolyQ;*ubql-1*(RNAi) (*n* = 36), $p = 0.9424$; iPolyQ EV vs *hsf-1*OE;iPolyQ EV, $p < 0.0001$; iPolyQ;*ubql-1*(RNAi) vs *hsf-1*OE; iPolyQ;*ubql-1*(RNAi), $p < 0.0001$, and (**g**) mPolyQ EV (*n* = 25) vs mPolyQ EV (*n* = 21) vs *hsf-1*OE;mPolyQ EV (*n* = 21) vs *hsf-1*OE;mPolyQ;*ubql-1*(RNAi) (*n* = 23), $p < 0.0001$; mPolyQ EV vs *hsf-1*OE;mPolyQ EV, $p < 0.0001$; mPolyQ;*ubql-1*(RNAi) vs *hsf-1*OE;mPolyQ;*ubql-1*(RNAi), $p < 0.0001$.
**h, i** western blotting for K48-linked ubiquitylated proteins on (**h**) day 1 or (**i**) day 5 of adulthood in wildtype, *ubql-1(tm1574)*, *hsf-1* OE, and *hsf-1* OE;*ubql-1(tm1574)* animals. Blots are representative of 4 independent experiments. All error bars denote SD. Statistical significance was calculated using (**a–c**, **f** and **g**) two-way ANOVA with Fishers LSD and (**d**) Mantel-Cox log-rank test. ns, not significant ($p > 0.05$), *$p < 0.05$, **$p < 0.01$, ***$p < 0.001$, ****$p < 0.0001$. Full statistics for lifespan trials (including n values) can be found in Supplementary Data 2. Source data are provided as a Source Data file.

Fig. 2h, i). These data suggest that both *ubql-1* and *hsf-1* OE suppress the accumulation and aggregation of K48-linked ubiquitylated proteins with age, but that *ubql-1* is not necessary for *hsf-1* OE to mediate these effects.

### *Ubql-1* is necessary for a sub-set of transcriptomic and proteomic changes induced by *hsf-1* overexpression

While we cannot completely rule out changes in proteostasis as a contributor to *hsf-1* OE-mediated lifespan extension, our data suggested that *ubql-1* may be influencing longevity in *hsf-1* OE animals through alternative mechanisms to general proteostasis maintenance. Therefore, we employed an unbiased transcriptomics and proteomics approach to identify the genes, proteins and processes that may facilitate lifespan extension downstream of *ubql-1* in *hsf-1* OE worms.

We found that HSF-1 overexpression generated substantial changes across both the proteome and transcriptome, with a total of 3262 transcripts (2343 increased and 919 decreased, (Log 2-FC, FDR *p* < 0.05)) and 1564 proteins (948 increased and 616 decreased, FDR < 0.05) altered compared to wildtype worms (Supplementary Fig.3a, b, Supplementary Data 3 and Supplementary Data 6). While we did not observe a strong overlap in the specific identities of altered proteins and genes across our proteomic and transcriptomic datasets (Supplementary Fig. 3a, b), there was a good functional correlation between the two, with KEGG analysis[24] revealing that up-regulated genes and proteins were enriched for pathways regulating metabolism and protein processing in the endoplasmic reticulum (ER), while down-regulated genes and proteins were enriched for pathways that included mismatch repair, DNA replication and signalling pathways (Supplementary Fig.3c, d, Supplementary Data 4 and Supplementary Data 7).

In contrast, loss of *ubql-1* resulted in fewer transcriptomic and proteomic changes in both wild-type and *hsf-1* OE worms, with *ubql-1(tm1574)* mutants exhibiting changes in 418 genes (402 up-regulated and 16 down-regulated (Log 2-FC, FDR *p* < 0.05) and 68 proteins (43 increased and 25 decreased (FDR < 0.05)) compared to wildtype worms (Supplementary Fig. 3e, f, Supplementary Data 3 and Supplementary Data 6). Loss of *ubql-1* altered the expression of 152 genes (75 up-regulated and 77 down-regulated (Log 2-FC, FDR *p* < 0.05)) and 43 proteins (23 increased and 20 decreased (FDR < 0.05)) in *hsf-1* OE worms (Fig. 3a, b, Supplementary Data 3 and Supplementary Data 6). Gene ontology analysis revealed that *ubql-1* dependent genes in *hsf-1* OE worms were enriched for the terms "cell projection organisation" (*p* < 0.00018) and "defence response" (*p* < 0.00022), while differentially regulated proteins were enriched for the term "actin-based filaments" (Supplementary Data 4 and Supplementary Data 7). The enrichment for "metabolic pathways" among *hsf-1* OE mediated changes in gene expression and protein abundance prompted us to use WormFlux[25] to investigate whether any metabolic pathways were significantly altered by loss of *ubql-1*. This revealed "Fatty acid biosynthesis" (ACS-2, PPT-1; P-enrichment = 0.00096) and "nicotine and nicotinamide metabolism" (*parg-2*; P-enrichment = 0.038) as pathways that are disrupted in *hsf-1* OE worms lacking *ubql-1* (Supplementary Data 4 and Supplementary Data 7). Finally, an assessment of subcellular localisation revealed that of the 43 proteins with altered abundance in *hsf-1* OE; *ubql-1* mutants, 8 were annotated as "integral membrane proteins" (Supplementary Data 3). These results indicate that *ubql-1* regulates the expression/abundance of genes/proteins associated with diverse processes, with membrane-associated and fatty-acid biosynthesis proteins affected.

Of the transcriptomic and proteomic changes observed in *hsf-1* OE worms compared to wild-type animals, 61 genes and 12 proteins were increased in a *ubql-1* dependent manner, and 17 genes and 4 proteins were down-regulated in a *ubql-1* dependent manner (Fig. 3c and d). Given that ubiquilins have a central role in protein degradation pathways associated with the cytosol/nucleus, endoplasmic reticulum (ER) and mitochondria[13,18,26,27], we reasoned that *ubql-1* primarily

promotes lifespan extension in *hsf-1* OE animals by promoting the degradation of key target proteins, with transcriptional changes arising as a secondary consequence of these effects. Despite this, we observed 12 proteins whose abundance decreased in *hsf-1* OE worms upon loss of *ubql-1* (Fig. 3d). Among these, was acyl-CoA synthetase-2 (ACS-2), a key enzyme in mitochondrial beta-fatty acid oxidation that is known to localise to mitochondria[28–33] (Fig. 3d). *Acs-2* expression is strongly controlled by the transcription factor NHR-49, which has been shown to be necessary for lipid homeostasis and increased lifespan, including in *hsf-1* OE worms[9,28–33]. Therefore, we hypothesised that *ubql-1* may be exerting a positive effect on lifespan by promoting NHR-49 activity.

As expected, we found that *nhr-49* (RNAi) shortens the lifespan of wild-type worms and prevents *hsf-1* OE-mediated lifespan extension (Supplementary Fig. 3g, h and Supplementary Data 2). However, the knockdown of *ubql-1* did not alter NHR-49 activity in *hsf-1* OE worms (Supplementary Fig. 3i, j), suggesting that *ubql-1* does not influence ACS-2 levels or lifespan in *hsf-1* OE animals through regulation of NHR-49 activity.

### Loss of *ubql-1* leads to the accumulation of NPL-4.1 in worms overexpressing *hsf-1*

As we hypothesised that UBQL-1 activity is increased upon *hsf-1* overexpression, we predicted that this would promote the degradation of UBQL-1 targets in *hsf-1* OE worms. We, therefore, reasoned that proteins whose levels were increased in *hsf-1* OE animals upon loss of *ubql-1*, may be UBQL-1 clients. Amongst these, the protein NPL-4.1 displayed the strongest increase in abundance in *hsf-1* OE; *ubql-1* mutant animals (Fig. 3d). NPL-4.1 is a central component of the CDC-48-NPL-4-UFD1 complex, which is at the core of both ER-associated protein degradation (ERAD) and mitochondria-associated protein degradation (MAD)[34]. In both contexts, the CDC-48 AAA-ATPase complex acts to "pull" ubiquitylated substrates free from organelle membranes[35]. In ERAD, this process is facilitated by NPL4 and UFD1, which bind to one side of the CDC-48 hexamer and interact with ubiquitin chains, thereby directing polyubiquitinated proteins to the CDC-48 complex for extraction[35]. While less well-studied, NPL4-UFD1 are also necessary for the role of CDC-48 in MAD, presumably through similar functions as in ERAD[36,37].

We observed a strong reduction in *npl-4.1*, *npl-4.2*, *ufd-1*, *cdc-48.1* and *cdc-48.2* mRNA levels in hsf-1 OE worms compared to wild-type animals, although transcript levels were not restored in *ubql-1* mutants (Fig. 3e). Unlike NPL-4.1, we did not observe an increase in the abundance of any other CDC-48 complex components in *hsf-1* OE; *ubql-1* mutants (Fig. 3f). Together, our data reveal that *hsf-1* OE promotes transcriptional down-regulation of genes encoding CDC-48 complex components, and a *ubql-1* dependent reduction in NPL-4.1, which could lead to impairment of organellar protein degradation.

### Overexpression of *hsf-1* is associated with sensitivity to ER stress and altered mitochondrial morphology

Given that the expression of genes encoding CDC-48 complex components is reduced in *hsf-1* OE animals, we hypothesised that lifespan extension could arise from hormetic effects associated with compromised organellar protein degradation. Consistent with a reduction in ERAD, we observed that "protein processing in the ER" was the strongest enriched term among our set of increased proteins in *hsf-1* OE worms (Supplementary Fig. 3d) and that *hsf-1* OE worms were sensitive to ER stress induced by tunicamycin (Fig. 4a). While we did not observe evidence for activation of the mitochondrial unfolded protein response (UPR[mt]) with age in *hsf-1* OE worms (Supplementary Fig.4b–d), we did find that mitochondrial network organisation was dramatically altered upon *hsf-1* OE, with *hsf-1* OE animals exhibiting a more fused mitochondrial network early in adulthood (Fig. 4b–d).

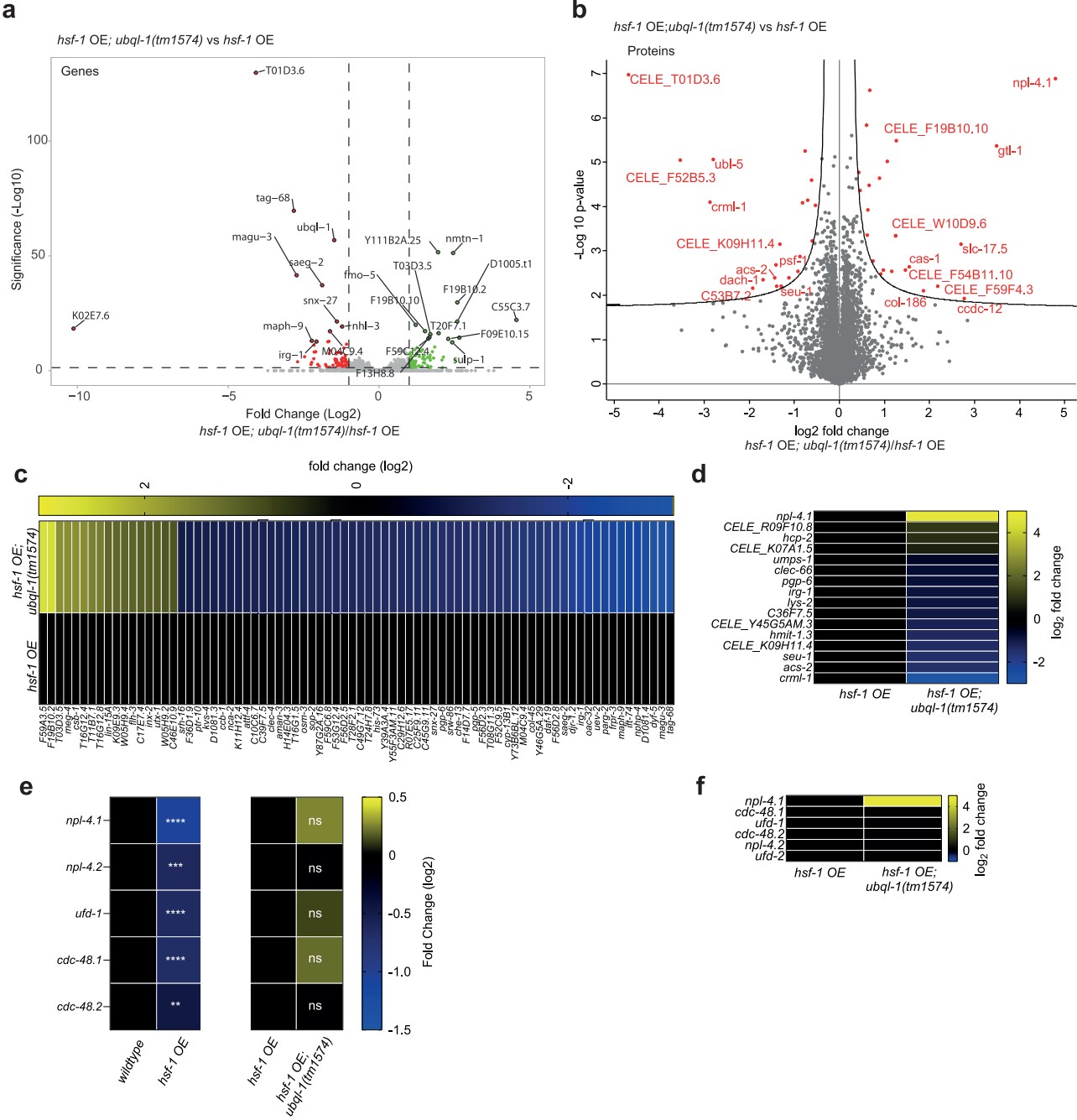

**Fig. 3 | Ubql-1 is necessary for elevated NPL-4.1 levels in hsf-1 OE animals.**
**a, b** Volcano plots representing differentially regulated (**a**) genes (green dots = up-regulated, red dots = down-regulated, FDR *p* < 0.05) and (**b**) proteins (red dots in the right and left region of the volcano plot represent proteins significantly up-or down-regulated, respectively (two-tailed student's *t* test, FDR < 0.05) in *hsf-1* OE *vs hsf-1* OE; *ubql-1(tm1574)* worms on day 1 of adulthood. Values are the mean of 4 biological replicates. **c, d** Heatmaps of *ubql-1* dependent changes in (**c**) gene

expression and (**d**) protein abundance in *hsf-1* OE worms. **e** Heatmap showing the relative expression of CDC-48-NPL-4-UFD-1 complex associated genes within RNA-seq datasets from wildtype vs *hsf-1* OE worms, and *hsf-1* OE vs *hsf-1* OE; *ubql-1(tm1574)* worms. **f** Heatmap showing the relative abundance of CDC-48-NPL-4-UFD-1 complex associated proteins within proteomics datasets from *hsf-1* OE vs *hsf-1* OE; *ubql-1(tm1574)* worms. Source data are provided as a Source Data file.

Of the 417 mitochondrial proteins identified in our proteomics dataset, 66 exhibited altered abundance in *hsf-1* OE animals (FDR *P* < 0.05), with 22 increased and 44 decreased (Supplementary Fig. 4a and Supplementary Data 5). Of these, 21 proteins are annotated as outer or inner membrane proteins. These data suggest that *hsf-1* OE alters the stability of a subset of the mitochondrial proteome and that the increased mitochondrial fusion in *hsf-1* OE worms is unlikely to be the result of a general change in mitochondrial mass.

Our data show that *hsf-1* OE animals exhibit ER and mitochondrial phenotypes that are consistent with changes in CDC-48-NPL-4-UFD-1 activity. To determine how gross defects in ERAD or MAD impact lifespan in *hsf-1* OE or wild-type worms, we used RNAi to knock down the E3 ligase, SEL-11/HRD-1 (which is necessary for ERAD)[38], or individual components of the CDC-48-NPL-4-UFD-1 complex. Knockdown of *sel-11* reduced lifespan to a comparable extent in both wildtype and *hsf-1* OE worms, suggesting that ERAD is

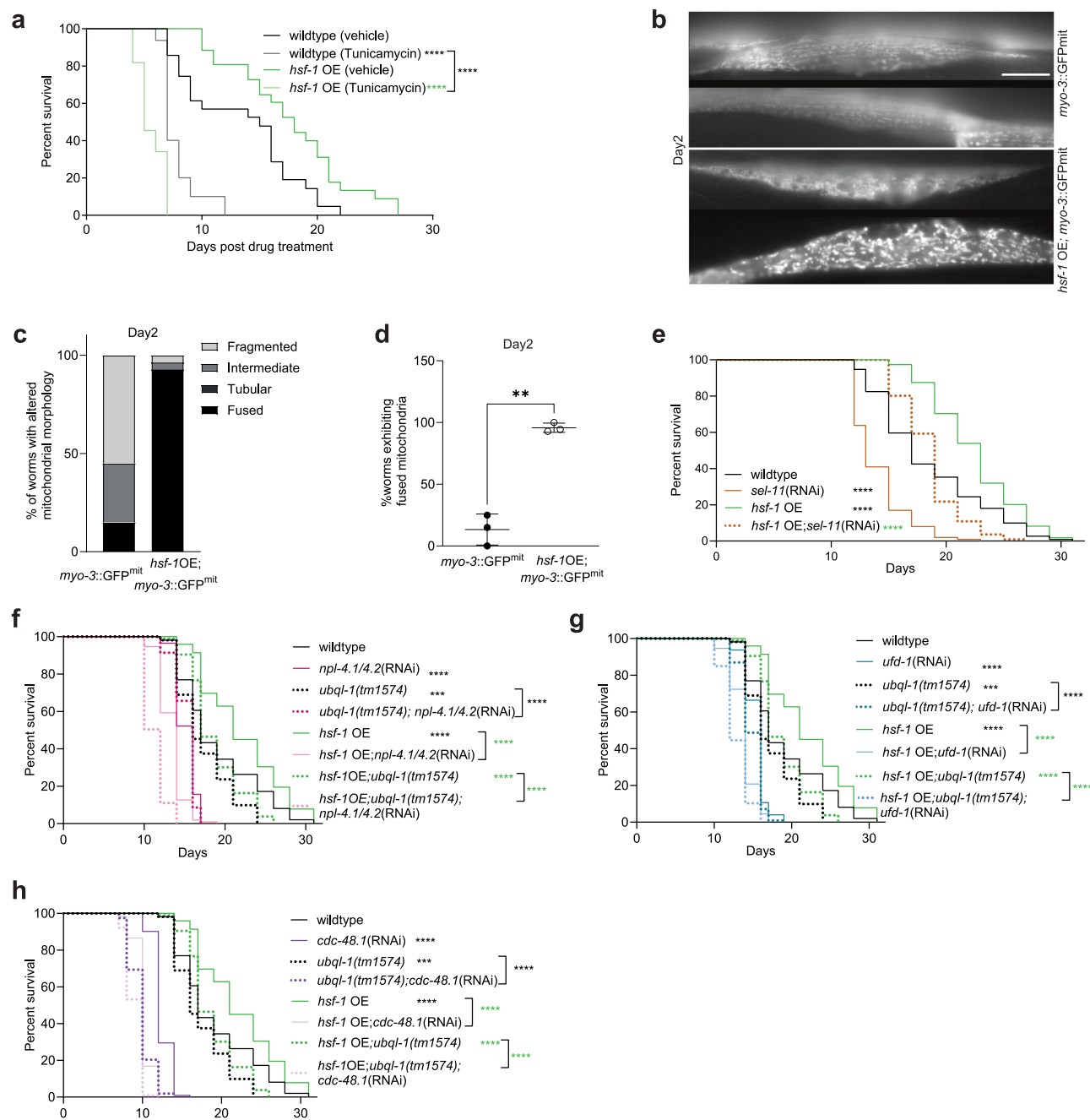

**Fig. 4 | Animals overexpressing HSF-1 exhibit altered mitochondrial morphology and are sensitive to knockdown of the CDC-48-UFD-1-NPL-4 complex.**
**a** Survival of wildtype and *hsf-1* OE worms exposed to Tunicamycin (50 μg/ml) or DMSO (0.2%) on NGM plates from day 1 of adulthood onwards (wildtype DMSO vs wildtype tunicamycin, *p* < 0.0001; *hsf-1* OE DMSO vs *hsf-1*OE tunicamycin, *p* < 0.0001; wildtype tunicamycin vs *hsf-1*OE tunicamycin, *p* < 0.0001).
**b** Representative fluorescence images of *myo-3*p::GFPmit in wildtype or *hsf-1* OE muscle tissues on day 2 of adulthood. Scale bar, 20 μm. **c** Proportion of mitochondria exhibiting specific morphologies in wildtype or *hsf-1* OE muscle tissues on day 2 of adulthood (*myo-3*::GFP(mit), *n* = 20; *hsf-1*OE; *myo-3*::GFP(mit), *n* = 28).
**d** Relative prevalence of fused mitochondria in wildtype or *hsf-1* OE muscle tissues on day 2 of adulthood. Data are presented as the mean values +/− SD of 3 independent experiments (*p* = 0.0046). This experiment was also run +/− *ubql-1* RNAi, which is plotted in Fig. S5b. **e** Lifespan analysis of wildtype and *hsf-1* OE animals subjected to *sel-11* RNAi (wildtype vs *sel-11*(RNAi), *p* < 0.0001; wildtype vs *hsf-1*OE, *p* < 0.0001; *hsf-1*OE vs *hsf-1*OE;*sel-11*(RNAi), *p* < 0.0001). **f**–**h** Lifespan analyses of wildtype, *ubql-1 (tm1574)* mutants, *hsf-1* OE and *hsf-1* OE;*ubql-1(tm1574)* animals subjected to empty vector pL4440 and RNAi-mediated (**f**) *npl-4.1/4.2*, (**g**) *ufd-1* and

(**h**) *cdc-48.1* knockdown (wildtype vs *npl-4.1/4.2*(RNAi), *p* < 0.0001; wildtype vs *ubql-1(tm1574)*, *p* = 0.0008; *ubql-1(tm1574)* vs *ubql-1(tm1574)*;*npl-4.1/4.2*(RNAi), *p* < 0.0001; wildtype vs *hsf-1*OE, *p* < 0.0001; *hsf-1*OE vs *hsf-1*OE;*npl-4.1/4.2*(RNAi), *p* < 0.0001; *hsf-1*OE vs *hsf-1*OE;*ubql-1(tm1574)*, *p* < 0.0001; *hsf-1*OE;*ubql-1(tm1574)* vs *hsf-1*OE; *ubql-1(tm1574); npl-4.1/4.2*(RNAi), *p* < 0.0001; wildtype vs *ufd-1*(RNAi), *p* < 0.0001; wildtype vs *ubql-1(tm1574)*, *p* = 0.0008; *ubql-1(tm1574)* vs *ubql-1(tm1574)*; *ufd-1*(RNAi), *p* < 0.0001, wildtype vs *hsf-1*OE, *p* < 0.0001; *hsf-1*OE vs *hsf-1*OE; *ufd-1*(RNAi), *p* < 0.0001; *hsf-1*OE vs *hsf-1*OE;*ubql-1(tm1574)*, *p* < 0.0001; *hsf-1*OE;*ubql-1 (tm1574)* vs *hsf-1*OE;*ubql-1(tm1574)*;*ufd-1*(RNAi), *p* < 0.0001; wildtype vs *cdc-48.1*(RNAi), *p* < 0.0001; wildtype vs *ubql-1(tm1574)*, *p* = 0.0008; *ubql-1(tm1574)* vs *ubql-1(tm1574)*;*cdc-48.1* (RNAi), *p* < 0.0001; wildtype vs *hsf-1*OE, *p* < 0.0001; *hsf-1*OE vs *hsf-1*OE;*cdc-48.1* (RNAi), *p* < 0.0001; *hsf-1*OE vs *hsf-1*OE;*ubql-1(tm1574)*, *p* < 0.0001; *hsf-1*OE;*ubql-1(tm1574)* vs *hsf-1*OE;*ubql-1(tm1574)*; *cdc-48.1* (RNAi), *p* < 0.0001). Statistical significance was calculated using (**d**) two-way ANOVA with Fishers LSD or (**a**, and **e**–**h**) Mantel-Cox log-rank test. **p* < 0.01, ***p* < 0.001, ****p* < 0.0001. Full statistics for lifespan trials (including n values) can be found in Supplementary Data 2. Source data are provided as a Source Data file.

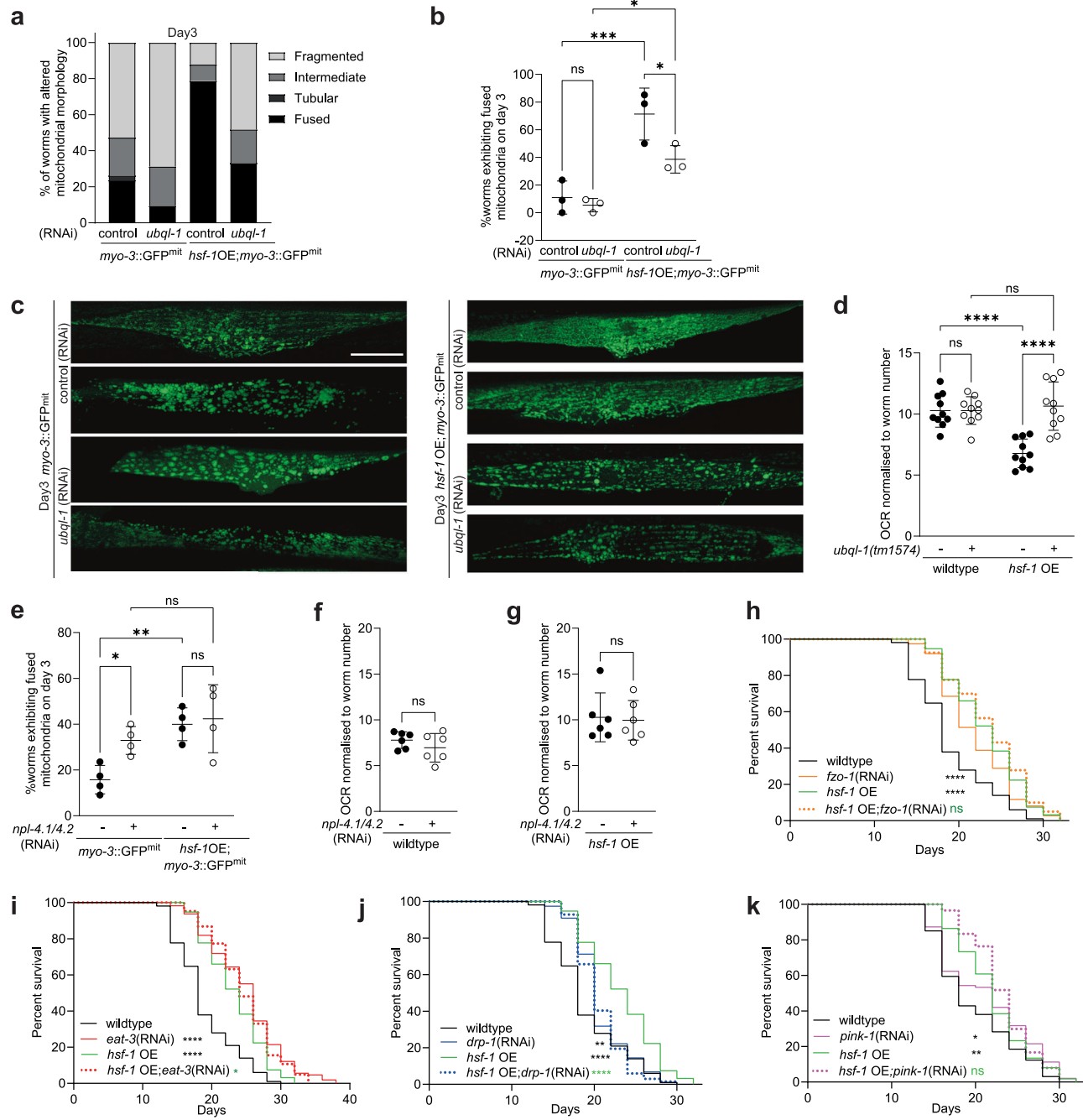

required for normal survival but does not interact with *hsf-1* OE to increase lifespan (Fig. 4e). In contrast, while knockdown of *npl-4.1/4.2*, *ufd-1* or *cdc-48.1* was detrimental to all genotypes tested, *hsf-1* OE worms were more sensitive to knockdown of these factors than wildtype worms (Fig. 4f–h).

In addition, the loss of *ubql-1* further shortened the lifespans of *hsf-1* OE mutants exposed to *npl-4.1*(RNAi). This likely reflects the fact that loss of *ubql-1* is unlikely to indefinitely maintain/increase NPL-4.1 protein levels in *hsf-1* OE worms in the face of prolonged RNAi against *npl-4.1*. As such, the absence of a fully functional *ubql-1*, combined with reduced expression of *cdc-48.1* and *ufd-1*, leaves *hsf-1* OE animals more vulnerable to *npl-4.1* knockdown. Together, these data suggest that HSF-1 overexpression reduces CDC-48-NPL-4-UFD-1 function and that this is accompanied by a more fused mitochondrial network and altered metabolic homeostasis.

## HSF-1 overexpression promotes lifespan by altering mitochondrial dynamics

To shed light on whether *ubql-1* is necessary for the changes in mitochondrial networks observed in *hsf-1* OE worms and whether this is associated with altered mitochondrial function and/or lifespan extension, we first assessed mitochondrial morphology in wildtype and *hsf-1* OE animals in the presence or absence of *ubql-1* (RNAi). Knockdown of *ubql-1* suppressed mitochondrial fusion in *hsf-1* OE, but not wild-type worms on days 2 and 3 of adulthood (Fig. 5a–c and Supplementary Fig. 5a–c). The abundance of mitochondrial proteins was largely unaffected in *hsf-1* OE; *ubql-1* mutants, with just one mitochondrial protein significantly increased (ACS-2) and none significantly decreased (Supplementary Fig. 5d and Supplementary Data 5). Furthermore, mitochondrial copy number was unaltered in *hsf-1* OE or *ubql-1* mutants (Supplementary Fig. 5e). Together, these data suggest that loss of *ubql-1* does not alter mitochondrial mass.

**Fig. 5 | hsf-1 overexpression reduces respiration and promotes mitochondrial fusion in a ubql-1 dependent manner. a** Proportion of mitochondrial morphologies observed in $p_{myo-3}$::GFP(mit) worms and *hsf-1* OE; $p_{myo-3}$::GFP(mit) worms grown on empty vector control or *ubql-1*(RNAi) on day 3 of adulthood (*myo-3*::GFP(mit), $n = 38$; *myo-3*::GFP(mit);*ubql-1*(RNAi), $n = 32$; *hsf-1*OE; *myo-3*::GFP(mit), $n = 33$; *hsf-1*OE; *myo-3*::GFP(mit); *ubql-1*(RNAi), $n = 27$. **b** Prevalence of fused mitochondria in muscle tissues of wildtype and *hsf-1* OE worms, +/− *ubql-1* (RNAi), on day 3 of adulthood. Values are the mean +/- SD of three independent experiments (*myo-3*::GFP(mit) vs *myo-3*::GFP(mit);*ubql-1*(RNAi), $p = 0.6018$; *hsf-1*OE; *myo-3*::GFP(mit) vs *hsf-1*OE; *myo-3*::GFP(mit); *ubql-1*(RNAi), p = 0.0120; *myo-3*::GFP(mit) vs *hsf-1*OE;*myo-3*::GFP(mit), $p = 0.0003$; *myo-3*::GFP(mit);*ubql-1*(RNAi) vs *hsf-1*OE; *myo-3*::GFP(mit);*ubql-1*(RNAi), $p = 0.0113$). **c** Representative confocal microscope images of *myo-3*p::GFPmit within muscle tissues of wildtype or *hsf-1* OE worms +/− *ubql-1*(RNAi) on day 3 of adulthood. Scale bar, 20 μm. **d** Basal Oxygen consumption rates (OCR) in wildtype, *ubql-1(tm1574)*, *hsf-1* OE and *hsf-1* OE; *ubql-1(tm1574)* animals at day 1 of adulthood. Mean values +/− SD ($n = 10$ replicates per group where each replicate has 10 worms) are plotted. One of three independent experiments has been represented for OCR (wildtype vs *ubql-1(tm1574)*, $p = 0.9947$; *hsf-1*OE vs *hsf-1*OE;*ubql-1(tm1574)*, $p < 0.0001$; wildtype vs *hsf-1*OE, $p < 0.0001$; *ubql-1 (tm1574)* vs *hsf-1*OE;*ubql-1(tm1574)*, $p = 0.5736$). **e** Prevalence of fused mitochondria in muscle tissues of wildtype and *hsf-1* OE worms, +/− *npl-4.1* (RNAi), on day 3 of adulthood. The plotted values are from four independent experiments. The number of worms imaged per experiment was: Experiment 1, $n = 10$ for all groups; Experiment 2, $n = 10$ for all groups except wildtype *npl-4.1*(RNAi) ($n = 5$); Experiment 3 and 4,

$n = 15$ for all groups. Statistical comparisons are: *myo-3*::GFP(mit) vs *myo-3*::GFP(mit);*npl-4.1/4.2*(RNAi), $p = 0.0230$; *hsf-1*OE; *myo-3*::GFP(mit) vs *hsf-1*OE; *myo-3*::GFP(mit); *npl-4.1/4.2*(RNAi), $p = 0.7201$; *myo-3*::GFP(mit) vs *hsf-1*OE;*myo-3*::GFP(mit), $p = 0.0032$; *myo-3*::GFP(mit);*npl-4.1/4.2*(RNAi) vs *hsf-1*OE; *myo-3*::GFP(mit); *npl-4.1/4.2*(RNAi), $p = 0.1766$. (**f** and **g**) Basal OCR in (**f**) wildtype or (**g**) *hsf-1* OE animals, +/− *npl-4.1/4.2* (RNAi), at day 1 of adulthood. Mean values are plotted +/− SD ($n = 6$ replicates per group where each replicate has 10 worms (wildtype vs wildtype;*npl-4.1/4.2*(RNAi), $p = 0.2869$; *hsf-1*OE vs *hsf-1*OE;*npl-4.1/4.2*(RNAi), $p = 0.8256$. One of two independent experiments has been shown. **h**–**k** Lifespan of wildtype and *hsf-1* OE animals on empty vector and (**h**) *fzo-1*(RNAi) (fusion) (wildtype vs wildtype;*fzo-1*(RNAi), $p < 0.0001$; wildtype vs *hsf-1*OE, $p < 0.0001$; *hsf-1*OE vs *hsf-1*OE;*fzo-1*(RNAi), $p = 0.3953$), (**i**) *eat-3* (RNAi) (fusion) (wildtype vs wildtype;*eat-3*(RNAi), $p < 0.0001$; wildtype vs *hsf-1*OE, $p < 0.0001$; *hsf-1*OE vs *hsf-1*OE;*eat-3*(RNAi), $p = 0.0134$) (**j**) *drp-1*(RNAi) (fission) (wildtype vs wildtype;*drp-1*(RNAi), $p = 0.0086$; wildtype vs *hsf-1*OE, $p < 0.0001$; *hsf-1*OE vs *hsf-1*OE;*drp-1*(RNAi), $p < 0.0001$) and (**k**) *pink-1* (RNAi) (mitophagy) (wildtype vs wildtype;*pink-1*(RNAi), $p = 0.0166$; wildtype vs *hsf-1*OE, $p = 0.0020$; *hsf-1*OE vs *hsf-1*OE;*pink-1*(RNAi), $p = 0.1268$). All error bars denote SD. Statistical significance was calculated by (**b**, **d**, and **e**) two-way ANOVA with Fishers LSD test, (**f**, **g**) two-tailed, unpaired Students *t* test, or (**h**–**k**) Mantel-Cox Log-rank test. ns, not significant ($p > 0.05$), *$p < 0.05$, **$p < 0.01$, ***$p < 0.001$ ****$p < 0.0001$. Full statistics for lifespan trials (including n values) can be found in Supplementary Data 2. Source data are provided as a Source Data file.

Overexpression of *hsf-1* also reduced oxygen consumption rates (OCR) in a *ubql-1* dependent manner but did not alter ATP levels or total triglyceride levels (measured by Oil-Red-O staining[39]) (Fig. 5d, and Supplementary Fig. 5f–h). However, we did observe a reduction in lipid stores upon loss of *ubql-1* in both wildtype and *hsf-1* OE worms (Supplementary Fig. 5g, h), suggesting a previously undiscovered role for ubiquilins in lipid homeostasis.

To understand whether the effects of *hsf-1* OE on mitochondrial fusion might be mediated by reduced NPL-4.1 levels, we exposed wildtype and *hsf-1* OE worms to *npl-4.1* (RNAi) and assessed mitochondrial morphology and OCR. We found that *npl-4.1* (RNAi) was sufficient to increase mitochondrial fusion in wild-type animals but did not further increase fusion in *hsf-1* OE animals (Fig. 5e). However, knockdown of *npl-4.1* was not sufficient to induce an accompanying change in OCR (Fig. 5f, g).

Lastly, we wanted to test whether mitochondrial network dynamics are important for the extended lifespan of *hsf-1* OE worms. Maintenance of mitochondrial networks is controlled via fission and fusion factors[40,41], therefore, we measured survival in worms in which core fusion/fission mediators were knocked down by RNAi, specifically, the cytosolic dynamin-like GTPase DRP-1 and the membrane-anchored dynamin-like GTPases FZO-1/mitofusin and EAT-3/OPA1[42–44].

RNAi against *fzo-1* or *eat-3* extended lifespan in wild-type worms to a level comparable to that seen upon *hsf-1* OE (Fig. 5h, i). However, no additive effect on lifespan was observed in *hsf-1* OE; *fzo-1*(RNAi) or *eat-3*(RNAi) groups (Fig. 5h, i). In contrast, *drp-1*(RNAi) only modestly increased lifespan in wild-type animals and strongly suppressed lifespan extension upon *hsf-1* OE (Fig. 5j). Given that mitochondrial fission contributes to mitochondrial homeostasis by segregating damaged mitochondria for degradation via mitophagy[45], we investigated whether mitophagy is also necessary for the elevated lifespan of *hsf-1* OE worms. Knockdown of the kinase PINK-1 (a core mitophagy regulator) did not suppress lifespan extension in *hsf-1* OE worms (Fig. 5k).

Together, our data suggest that in *hsf-1* OE animals, increased *ubql-1* expression reduces NPL-4.1 levels, which promotes longevity by altering mitochondrial network dynamics. We propose that the alterations in mitochondrial morphology that we observe in *hsf-1* OE worms may be the result of reduced organellar protein degradation.

## Discussion

Our work identifies *ubql-1* as a key determinant of the extended longevity conferred by overexpression of HSF-1. We propose that ubiquilin-1 promotes the long lifespan of *hsf-1* OE animals by facilitating mitochondrial fusion rather than broadly altering proteostasis capacity. Interestingly, a recent study linked activation of the UPR^mt to the increased lifespan of *C. elegans hsb-1* mutants (which also have increased HSF-1 activity), suggesting that altered mitochondrial homeostasis may be a general feature of the beneficial effects conferred by enhancing HSF-1 activity[46].

Research in yeast and mammalian cells has shown that ubiquilin proteins localise to the nucleus and cytoplasm and function in protein degradation pathways by acting as shuttle proteins that transport ubiquitinated substrates to both the proteasome and the autophagy-lysosomal systems[12,47]. Ubiquilins are also known components of ERAD complexes[48], and the knockdown of *ubql-1* in worms leads to induction of the ER unfolded protein response and the accumulation of misfolded proteins[26,49]. Consistent with this, we did observe that levels of K48-linked ubiquitylated proteins were increased in *ubql-1* mutant worms, but we did not see a large increase in polyubiquitylated proteins as was previously reported with *ubql-1*(RNAi)[26]. This likely reflects differences in acute versus chronic loss of UBQL-1 function and the life stage at which worms were examined[26].

UBQLN1 has also previously been linked to the maintenance of mitochondrial protein homeostasis, with studies showing that UBQLN1 targets mislocalized mitochondrial outer membrane proteins for proteasomal degradation in human cells. In addition, UBQLN1-deficient cells accumulate mislocalized mitochondrial proteins in the cytosol, leading to a loss in viability[18,26]. It is perhaps surprising then that we did not identify a greater number of proteins exhibiting altered abundance in wildtype or *hsf-1* OE worms upon loss of *ubql-1*. One possible explanation for this is that there may be residual UBQL-1 activity remaining in our mutant strain, as a truncated protein is produced rather than a complete knockout. Alternatively, UBQL-1 may only be critical for the turnover of a limited subset of proteins or the activity of other pathways, such as the autophagy-lysosome system, may be increased to compensate for the chronic loss of UBQL-1 activity in our mutants. Lastly, technical aspects, such as not using cytosolic fractions, or performing proteomics on whole animals, may also have masked changes.

Nevertheless, we were able to identify key proteins whose abundance was increased upon loss of UBQL-1 function, most notably, the NPL-4.1 subunit of the CDC-48-NPL-4-UFD complex. Our findings support a model in which the mild down-tuning of this complex leads to cellular adaptations that are beneficial for lifespan. Interestingly, *cdc-48.1* mutants have been shown to increase *C. elegans* lifespan by 50% when accompanied by loss of the deubiquitylase, *atx-3/ATXN3*[50], suggesting that down-tuning of CDC-48 activity can be beneficial. Future work to understand exactly when, where and by how much CDC-48.1 activity needs to be reduced to extend lifespan, will reveal the contexts in which inhibition of CDC-48 and its co-factors might be harnessed for beneficial effects.

We also identified several proteins whose abundance decreased in *hsf-1* OE worms upon loss of *ubql-1*. Among the most reduced proteins was the mitochondrial acetyl CoA-synthetase, ACS-2, which has a core role in fatty-acid metabolism and whose expression is controlled by the transcription factor, NHR-49[9,51]. Given that *nhr-49* and altered lipid metabolism are necessary for the increased lifespan of *C. elegans* possessing enhanced HSF-1 activity[52,53], it is interesting that we did not see any effect of *ubql-1* RNAi on NHR-49 activity. However, we did find that *ubql-1* was necessary for normal fat levels and elevated ACS-2 levels (a key target of NHR-49) in *hsf-1* OE worms. ACS-2 plays an important role in regulating fatty acid metabolism but is not required for normal lifespan[51]. However, ACS-2 does mediate lifespan extension in response to lysosomal signalling in *C. elegans*[51]. Future work to understand how, if at all, changes in ACS-2 and other *ubql-1* dependent metabolic processes impact *hsf-1* OE lifespan, will reveal whether these changes are causal to lifespan extension.

We observed that UBQL-1 promotes changes in mitochondrial network structure in *hsf-1* OE worms, with a more fused state becoming prevalent during early adulthood, and that manipulating either fission or fusion prevents *hsf-1* OE from extending lifespan. These observations are consistent with reports in dietary-restricted worms, where a shift in the balance of mitochondrial network homeostasis is more important for lifespan extension than specifically driving mitochondria towards a fully fused or fragmented state[41,54]. Future work to understand how long a preferentially fused state perseveres upon hsf-1 OE, as well as the degree of dynamism present within the system, will be important in fully understanding the relationship between *hsf-1* OE, mitochondrial dynamics and longevity.

The fact that worms overexpressing HSF-1 have low basal respiration despite a more fused mitochondrial network was unexpected, as mitochondrial fusion has previously been linked to higher OCR in mammalian cells[54–57]. However, studies in flies have indicated that the morphology and function of mitochondria can be dissociated[58]. Alternatively, it may be that a more longitudinal investigation of OCR and mitochondrial dynamics will reveal that OCR is restored or increased in *hsf-1* OE animals later in life. Understanding the precise nature of this relationship will help unravel the specific events that lead to lifespan extension when HSF-1 activity is increased.

Finally, while beyond the scope of this study to explore, it is also possible that the effects of ubiquilin-1 on mitochondria stem from interactions with the cytoskeleton. Ubiquilin-1 is important for cytoskeleton organisation in cancer cells[59], and mammalian ubiquilins were first identified as interactors of vimentin[60]. We observed that RHO GTPase-activating proteins, namely *rga-3* (K09H11.3) and *crml-1*, were increased in *hsf-1* OE worms in a *ubql-1* dependent manner. RHO GTPases have roles in actin cytoskeleton organisation[61], the maintenance of which has been shown to be modulated by HSF-1 in regulating ageing and stress resistance[62]. In yeast it is known that organisation of the cytoskeleton impacts mitochondrial quality control[63,64] and that mitochondrial dynamics are modulated by cytoskeletal interactions[65–67]. Therefore, future experiments to understand the interplay between ubiquilin activity, mitochondrial fission–fusion dynamics, cytoskeletal integrity and ageing may help expand our understanding of how HSF-1 promotes longevity.

In summary, our work establishes an HSF-1-UBQL-1-Mitochondrial axis that promotes longevity by remodelling mitochondrial networks in *C. elegans*. These findings challenge existing assumptions and highlight mitochondrial dynamics as an additional component of HSF-1-mediated lifespan extension.

## Methods

### *C. elegans* strains and culture conditions
All strains were maintained at 20 °C on NGM plates seeded with *Escherichia coli* OP50 under standard conditions[68] unless mentioned otherwise. Strains used in this study were: N2 Bristol (wild type laboratory strain), JPL52 *ubql-1(tm1574)4x*, AGD710 *uthIs235 [sur-5p::hsf-1::unc-54 3'UTR + myo-2p::tdTomato::unc-54 3' UTR]* referred to as *hsf-1* OE in this manuscript, JPL54 - AGD710 crossed to JPL52, AM140 *(rmIs132 [unc-54p::Q(35)::YFP])*, JPL66 - AGD710 crossed to AM140, AM738 *(rmIs297 [vha-6p::Q(44)::YFP, rol-6(su1006)])*, JPL67 - AGD710 crossed to AM738, SJ4103 *zcls14 [myo-3::GFP(mit)]*, JPL84 - AGD710 crossed to SJ4103, AM583 *myo-2p::gfp; hsf-1p::hsf-1*, WBM321 *wbmIs321[acs-2p::gfp + rol-6(su1006)]*, JPL88 *hsf-1 OE; wbmIs321[acs-2p::gfp + rol-6(1006)]*.

### RNA interference
All clones were sequenced and verified before use and were obtained from the Ahringer RNAi library[69]. RNAi was initiated by growing bacteria for 16 h at 37 °C in LB containing 100 µg/ml ampicillin, with shaking (24 x g). Cultures were then induced with 5 mM IPTG and allowed to grow at 37 °C for a further 3 h. After induction, bacteria were allowed to cool at room temperature and were then seeded onto NGM RNAi plates containing 100 µg/ml ampicillin and 1 mM IPTG. Plates with seeded bacteria were allowed to dry at room temperature before use.

### Screening for modifiers of lifespan in *hsf-1* OE worms
RNAi screening was carried out in flat-bottomed 12-well polystyrene plates, with wells supplemented with 1 ml of standard NGM media containing 1 mM IPTG and 100 µg/ml ampicillin. Each well was seeded with 100 µl of an individual RNAi culture (prepared as described above) and allowed to dry at room temperature. Once dry, FUdR was added to each well to a final concentration of 15 µM and also allowed to dry. N2 and *hsf-1* OE worms were stage-synchronised by hypochlorite treatment as described[68] and allowed to mature to the L4 larval stage on standard NGM plates. L4 worms were then washed off NGM plates and added to lifespan screening plates at a density of 20–30 per well. Worms were then scored every day for survival, with each RNAi screening plate run in triplicate for each genotype. Rare examples of progeny that escaped FUdR treatment or worms that exhibited vulval prolapse or internal hatching were removed from plates and censored. Survival was scored by gently touching individual worms with a platinum pick. Worms were scored as dead in the absence of pharyngeal pumping or a response to touching. All RNAi clones screened, and corresponding survival outcomes can be found in Supplementary Data 1.

### Lifespan, stress resistance and motility assays
Survival for lifespan and stress resistance assays were scored by gently touching/prodding worms with a platinum pick at the indicated time points. Worms were scored as dead when they stopped responding to prodding and in the absence of pharyngeal pumping. Censored worms in lifespan assays included exploded animals, those exhibiting bagging (internal hatching of progeny), rupturing through the vulva and worms that dried out on the edge of the plates and/or crawled off the plate. In stress resistance assays, worms were heat shocked on seeded NGM

plates at 33 °C for 4 h and allowed to recover at 20 °C. Censored worms in stress resistance assays were only those worms that dried out on the edge of the plates/crawled off the plate. For tunicamycin treatment, worms were transferred to seeded NGM plates containing tunicamycin (50 μg/ml) on day 1 of adulthood. Survival statistics for all experimental repeats can be found in Supplementary Data 2.

## Proteostasis sensor assays
Polyglutamine aggregation was scored in muscle and intestinal proteostasis sensors – AM140, JPL66, AM738, JPL67 grown on empty vector and *ubql-1*(RNAi) at indicated time points. For obtaining images and aggregate counting, worms were immobilised on 3% agar pads in 3 mM levamisole, and images were captured using a Zeiss Imager.Z2 microscope (10x) and Hamamatsu ORCA-Flash 4.0 digital CMOS camera. Aggregates were determined to be any discrete foci exhibiting fluorescence signal above the background diffuse signal.

## RNA extraction, cDNA synthesis and RTqPCR
Approximately 100–200 adult animals per treatment group were lysed in 250 ul of Trizol by vortexing for 10 minutes at 4 °C in three cycles. RNA was purified using an RNeasy extraction kit as per the manufacturer's instructions. cDNA was generated using 1 μg of total RNA and an iScript cDNA synthesis kit. Real-time quantitative PCR was performed using a Biorad CFX96 Real-time PCR detection system with CFX Manager v3.1 and BioRad SsoAdvanced SYBR green super mix. Expression of genes of interest was calculated relative to the housekeeping genes *pmp-3, rpb-2* and *cdc-42* using the standard curve method. Sequences for all primer pairs used in this study can be found in Supplementary Table 1.

## RNA-sequencing and analysis
RNA integrity was assessed using an RNA Nano 6000 assay kit and an Agilent Bioanalyzer 2100 system. Following this, mRNA was purified from 1 μg of total RNA using poly-dT magnetic beads and cDNA libraries were generated using random hexamers and M-MuLV reverse transcriptase for first-strand synthesis, followed by second-strand synthesis using DNA polymerase I and RNase H. Fragments were blunt-ended and adaptors were ligated before PCR was performed using Phusion high fidelity DNA polymerase. PCR products were purified using an AMPure XP system, and library quality was checked using an Agilent Bioanalyzer 2100. Libraries were sequenced using an Illumina Novaseq 6000 platform to generate paired-end 150 base pair reads at a depth of 20 million reads per sample. Following sequencing, raw data was checked and reads containing adaptor sequences, poly-N reads or poor-quality sequences were removed. Reads were then aligned to the *C. elegans* reference genome using Hisat2 v2.0.5, and featureCounts v1.5.0-p3 was used to quantify the number of reads mapped to each gene. Differential expression testing was carried out using DESeq2, and *p*-values were adjusted using Benjamini-Hochberg. Up or down-regulated genes were analysed using the g:Profiler tool[24] (biit.cs.ut.ee/gprofiler/gost) to identify KEGG processes/pathways that were enriched in the groups being compared. Volcano Plots were created using the web app, VolcaNoseR[70]. Raw and unprocessed data sets can be obtained through the Gene Expression Omnibus using accession number: GSE241558.

## Oxygen consumption assays
Oxygen consumption rate (OCR) was measured in worms using the Seahorse XF96 (Seahorse Bioscience) with Wave software, using established protocols[71]. Wildtype, *ubql-1(tm1574)*, *hsf-1* OE and *hsf-1* OE; *ubql-1(tm1574)* animals were cultured on NGM plates with OP50 bacteria and grown until day 1 (YA). For RNAi experiments, wildtype and *hsf-1* OE animals were cultured on plates with empty vector control or *npl-4.1/4.2* (RNAi) bacteria and grown until day 1 (YA). The evening before the experiment, the seahorse instrument was switched on to allow hydration of the instrument's probes prior to taking measurements. For this, an unused flux package (green lid; XFe96 extracellular flux assay kit) was used. The green sensor cartridge was removed and placed upside down so that the probe surface would not be scratched. 200 μl of Seahorse Bioscience XF96 calibrant (pH 7.4) solution was added to each well of the utility plate. The sensor cartridge was then placed back on the utility plate so that the probes dip into the wells. After placing the lid on top, the cartridge was incubated overnight at 37 °C without $CO_2$. On the day of analysis, day 1 (YA) worms were transferred from respective NGM/RNAi plates to plates without food to remove bacteria. In the meantime, the heater and temperature control were turned off on the XF software and background correction wells were marked. 22 μl of diluted FCCP (100 μM) and 24 μl of sodium azide (400 mM) were pipetted into the A and B injection ports of the hydrated utility plate with the sensor cartridge and calibrated on the seahorse instrument. Next, worms were picked to the 96-well Seahorse microplate. 8 oxygen consumption measurements were taken for determination of basal OCR. After this, drugs were injected – 9 and 4 cycles of measurements were taken after the addition of FCCP and sodium azide, respectively. Each cycle is: 2 min mix - wait for 30 secs -measure for 2 min. Data was normalised to worms per well. FCCP was made in DMSO. Sodium azide was prepared in distilled water.

## Mitochondrial morphology assays
For mitochondrial morphology imaging, the transgenic strains expressing GFP-tagged mitochondrial protein in the body wall muscles, *zcls14[myo-3::GFP(mit)]* strain and *hsf-1* OE;*zcls14[myo-3::GFP(mit)]* animals grown on empty vector and *ubql-1*(RNAi) were utilised. Worms were immobilised on 3% agar pads in 3 mM levamisole, and images were captured using either a Zeiss Imager.Z2 microscope and Hamamatsu ORCA-Flash 4.0 digital CMOS camera (63x/1.40 oil objective lens), or a Zeiss 980 Airyscan upright point scanning confocal microscope, equipped with C Plan-Apochromat 63x/1.40 oil objective on days 2 and 3 of adulthood. GFP fluorescence was excited with the 488 nm laser and acquired with a GaAsP-PMT detector (509 nm max emission) controlled with ZEN Blue software. Z-series optical sections were collected with a step-size of 1 micron, using the piezo. For experiments with *npl-4.1/4.2*(RNAi), all images were acquired on day 3 of adulthood using the Zeiss 980 Airyscan upright point scanning confocal microscope. While GFPmit is expressed in all body wall muscle cells, the region between the pharynx and vulva or the vulva and tail were selected for viewing muscle mitochondrial networks as these areas are less sensitive to mitochondrial fragmentation during mounting. Images were processed using ImageJ and ZEN 3.4 (blue edition). Qualitative analysis of the mitochondria morphology was done, and images were scored as fused, tubular, intermediate or fragmented based on the mitochondrial network organisation.

## Oil Red O staining
Staining for lipid levels was performed according to established protocols[39]. Wildtype, *ubql-1(tm1574)*, *hsf-1* OE and *hsf-1* OE*; ubql-1(tm1574)* worms were grown on NGM plates with OP50 bacteria to day 5 of adulthood and then harvested by washing plates 2-3 times with 200 μl of M9 buffer. Worms were transferred to 1.5 mL Eppendorf tubes, allowed to settle to the bottom and then washed a further two times by resuspension in 200 μl of M9 buffer to remove residual bacteria. Finally, 175 μL of M9 buffer was aspirated without disturbing the worms, and 200 μl of high-quality 60% isopropanol was added immediately. Worms were allowed to settle to the bottom of the tube before 175 μL of 60% isopropanol was removed and replaced with 200 μl of freshly filtered ORO working solution. Tubes were sealed with parafilm, and worms were stained for 18 h (overnight) at 25 °C in a wet chamber (wet paper towels in a plastic box). Oil Red O solution was then washed out using M9-0.01% Triton-X, and worms were mounted on 3% agarose pads. Images were acquired on a Nikon SMZ1270 stereo

microscope with a DS-Fi3 5.9 MP colour camera. Images were processed using ImageJ.

## Measuring NHR-49 activity
WBM321 (*wbmIs321[acs-2p::gfp + rol-6(su1006)]*) and JPL88 (*hsf-1* OE; *wbmIs321[acs-2p::gfp + rol-6(su1006)]*) strains were grown on empty vector, *ubql-1* or *nhr-49* RNAi until day 1 of adulthood. Worms were then immobilised on 3% agarose pads in 3 mM levamisole and imaged using a Zeiss Axioimage.Z2 microscope and Hamamatsu ORCA-Flash 4.0 digital CMOS camera (10x objective).

## ATP assays
Approximately 1000 wildtype, *ubql-1(tm1574)*, *hsf-1* OE and *hsf-1* OE;*ubql-1(tm1574)* animals were cultured on plates with OP50 bacteria and collected on Day 1 of adulthood (YA). Worm pellets were resuspended in 50 μl RIPA buffer lysate, and protein concentration in each sample was estimated using the PierceTM BCA Protein Assay Kit (Thermo Scientific) following the instructions of the manufacturer. ATP levels were quantified using a luciferin/luciferase-based ADP/ATP Ratio assay kit (Sigma Aldrich) as per the manufacturer's instructions. Luminescence was then measured using a Tecan Infinite M200 microplate reader with Magellan software, and levels of ATP were calculated.

## Protein extraction and western blotting
To extract protein for western blotting, wildtype, *ubql-1(tm1574)*, *hsf-1* OE and *hsf-1* OE; *ubql-1(tm1574)* worms (approximately 1000) were collected in M9, pelleted, and then resuspended in RIPA buffer supplemented with a protease inhibitor cocktail tablet (Four independent replicates were collected for each background on the days indicated). Worm pellets were flash-frozen in liquid nitrogen and crushed (4 cycles) in microcentrifuge tubes using a plastic dounce homogeniser. Lysates were then centrifuged at $24,000 \times g$ at 4 °C for 15 min, and the supernatant was collected. Protein concentration in each sample was estimated using the PierceTM BCA Protein Assay Kit (Thermo Scientific) following the instructions of the manufacturer. Proteins were separated by SDS-PAGE and transferred to nitrocellulose membranes before probing with primary antibodies. Blots were incubated with primary antibody for 1.5 h at room temperature (anti-HSP-6, a kind gift from the Morimoto lab, 1:1000; anti- α-tubulin, Sigma, T5168, clone B512, Lot 0000286919, 1:10000), or overnight at 4 °C in PBS-0.02% Tween-0.5% milk powder with gentle agitation (anti-PolyUb, Life Sensors, VU101, clone VU1, Lot AB-43653.001, 1:1000; anti-K48-Ub, Abcam, ab140601, clone EP8589, Lot GR3360615-1, 1:1000; anti-K63-Ub, Millipore, 05-1308, clone apu3, Lot 3638887,1:1000). Following incubation, blots were washed 3x with PBS-0.2% Tween, incubated with secondary antibodies for 1 h at room temperature (Invitrogen goat anti-rabbit HRP conjugated, Cat. 31463, (1:5000) or Invitrogen goat anti-mouse HRP conjugated, Cat. 31430, (1:5000)), washed a further 3x with PBS-0.2% Tween, and then developed using Biorad Clarity ECL detection reagents and Amersham ImageQuant800 or Odyssey XF Imager (HSP-6 blots only). Densitometry of protein bands was performed using ImageJ gel analysis tools.

## Proteomics
Approximately 1000 wildtype, *ubql-1(tm1574)*, *hsf-1* OE or *hsf-1* OE;*ubql-1(tm1574)* animals were cultured on plates with OP50 bacteria and collected on Day 1 of adulthood (YA). A total of 16 samples encompassing 4 biological replicates of each genotype, were collected and processed. Worm pellets were resuspended in 50ul RIPA buffer lysate (2%SDS) supplemented with a protease inhibitor cocktail tablet. Worm pellets were flash-frozen in liquid nitrogen and crushed (4 cycles) in microcentrifuge tubes using a plastic dounce homogeniser. Lysates were then heated at 95 °C for 10 min. Protein concentration in each sample was estimated using the PierceTM BCA Protein Assay Kit (Thermo Scientific) following the instructions of the manufacturer.

The collected worm lysate was boiled in lysis buffer (5% sodium dodecyl sulfate (SDS), 5 mm tris(2-carboxyethyl)phosphine (TCEP), 10 mm chloroacetamide (CAA), 100 mm Tris, pH 8.5) for 10 min followed by micro tip probe sonication (Q705 Sonicator from Fisherbrand) for 2 min with pulses of 1 s on and 1 s off at 80% amplitude. Protein concentration was estimated by NanoDrop (Thermo Fisher Scientific). Protein digestion was automated on a KingFisher APEX robot (Thermo Fisher Scientific) in a 96-well format using a protocol from Bekker-Jensen et al.[72] with some modifications. The 96-well comb is stored in plate #1, the sample in plate #2 in a final concentration of 70% acetonitrile and with magnetic MagReSyn Hydroxyl beads (ReSyn Biosciences) in a protein/bead ratio of 1:2. Washing solutions are in plates #3–5 (95% Acetonitrile (ACN)) and plates #6–7 (70% Ethanol). Plate #8 contains 300 μL digestion solution of 100 mm Tris pH 8.5 and trypsin (Promega) in an enzyme:protein ratio of 1:100. The protein aggregation was carried out in two steps of 1 min mixing at medium mixing speed, followed by a 10 min pause each. The sequential washes were performed in 2.5 min and slow speed, without releasing the beads from the magnet. The digestion was set to 12 h at 37 degrees with slow speed. Protease activity was quenched by acidification with trifluoroacetic acid (TFA) to a final pH of 2, and the resulting peptide mixture was purified on the OASIS HLB 96 wellplate (Waters). Peptides were eluted twice with 100 μL of 50% ACN and dried in a Savant DNA120 (Thermo Fisher Scientific).

Peptides were then dissolved in 2% formic acid before liquid chromatography-tandem mass spectrometry (MS/MS) analysis. The mixture of tryptic peptides was analysed using an Ultimate3000 high-performance liquid chromatography system coupled online to an Eclipse mass spectrometer (Thermo Fisher Scientific). Buffer A consisted of water acidified with 0.1% formic acid, while buffer B was 80% acetonitrile and 20% water with 0.1% formic acid. The peptides were first trapped for 1 min at 30 μl/min with 100% buffer A on a trap (0.3 mm by 5 mm with PepMap C18, 5 μm, 100 Å; Thermo Fisher Scientific); after trapping, the peptides were separated by a 50-cm analytical column (Acclaim PepMap, 3 μm; Thermo Fisher Scientific). The gradient was 7 to 35% B in 103 min at 300 nl/min. Buffer B was then raised to 55% in 3 min and increased to 99% for the cleaning step. Peptides were ionised using a spray voltage of 2.1 kV and a capillary heated at 280 °C. The mass spectrometer was set to acquire full-scan MS spectra (350–1400 mass/charge ratio) for a maximum injection time set to Auto at a mass resolution of 60,000 and an automated gain control (AGC) target value of 100%. For MSMS fragmentation, we chose the DIA approach: AGC target value for fragment spectra was set at 200%. 60 windows of 10 Da were used with an overlap of 1 Da (m/z range from 380 to 980). Resolution was set to 15,000 and IT to 40 ms. The normalised collision energy was set at 30%. All raw files were analysed by DIA-NN v1.8.1[73], searching against library generated automatically using *C. elegans* proteome (downloaded from UniProt) and standard settings: peptides from 7 to 30 AA, max number of missed cleavages of 1, oxidation (M) and protein-Acetylation as only variable modifications.

The data analysis was performed using the Perseus software platform. Briefly, the original data was first log2 transformed, and then only the proteins with at least 3 values from the 4 reps were kept. At this point, the missing data were imputed using the automatic settings of Perseus. The imputed data was then analysed on the Perseus platform, which calculated the difference in the intensity of each protein between the groups being compared and the FDR P-value (following Student's *t* tests). Volcano plots were then generated in Perseus by viewing and comparing the differential protein expression and FDR p-value between the genetic backgrounds being compared. A FDR p-value of 0.05 was applied as a cut-off (represented by black curves on each plot), and for each protein, significance is shown as -Log$_{10}$(p-value) coming from the t-test, as the difference between backgrounds compared.

The mass spectrometry proteomics data have been deposited to the ProteomeXchange Consortium via the PRIDE[74] partner repository with the dataset identifier PXD044595 [http://proteomecentral. proteomexchange.org/cgi/G etDataset?ID = PXD044595].

## Relative mtDNA content

mtDNA content relative to nuclear DNA in worms was quantified using 7 single Day 1 (YA) worms from wildtype, *ubql-1(tm1574)*, *hsf-1* OE and *hsf-1* OE;*ubql-1(tm1574)* worm populations cultured on OP50 bacteria. Each single worm was added to $10\,\mu l$ worm lysis buffer containing proteinase K and flash-frozen in liquid nitrogen. For standards, 5 Day1 (YA) worms from each background were collected in the same way as described earlier. All samples were incubated at $65\,°C$ for 90 minutes, then at $95\,°C$ for 15 min to release genomic DNA. Primers for nuclear gene *cdc-42p* and mitochondrial gene *nd-1* were used to estimate nuclear and mitochondrial DNA content. Quantitative PCR was performed using Biorad CFX96 Real-time PCR detection system and BioRad SsoAdvanced SYBR green super mix. Quantification of relative gene expression of mtDNA:nuclearDNA was performed with the standard curve method.

## Quantification and statistical analysis

Sample number (n) corresponds to the number of biological replicates/trials or the number of animals used, as stated in each figure legend. In all cases, error bars correspond to SD. All statistical tests (log-rank (Mantel-Cox), two-way ANOVA) were carried out as stated within each figure legend using GraphPad Prism 10. The statistical details of all experiments can be found in the accompanying figure legends.

## Reporting summary

Further information on research design is available in the Nature Portfolio Reporting Summary linked to this article.

## Data availability

The ChIP-seq datasets analysed during the study can be found in the NCBI Gene Expression Omnibus using accession number GSE81523. The RNA-seq data generated in this study have been deposited in the NCBI Gene Expression Omnibus using accession number GSE241558. The Proteomics data generated in this study have been deposited in the PRIDE repository with the accession number PXD044595 [http://proteomecentral.proteomexchange. org/cgi/G etDataset?ID = PXD044595]. Source data are provided in this paper.

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

## Acknowledgements

We thank members of the Labbadia lab and UCL Institute of Healthy Ageing for helpful discussions regarding the manuscript. We also thank the UCL Biosciences Molecular Biology Facility for the use of core equipment, Novogene UK for cDNA library preparation and RNA-sequencing, Prof. Rick Morimoto for sharing strains and the HSP-6 antibody, and Prof Will Mair for sharing the acs-2p::gfp strain. Some strains were provided by the CGC, which is funded by NIH Office of Research Infrastructure Programmes (P40 OD010440), and by the National Bioresource Project of Japan. J.L., X.W., R.W. and A.P.E. were funded by BBSRC grant BB/T013273/1. K.T. was funded by a Wellcome Collaborative Award in Science (209250/Z/17/Z). The Mass spectrometry instrumentation was funded by a Wellcome Trust Multiuser Equipment grant (221521/Z/20/Z) to K.T.

## Author contributions

A.P.E., X.W. and R.W. performed all experiments with the exception of mass-spectrometry based proteomics, which were performed by R.Z.C. All data were analysed by A.P.E. and R.Z.C. with supervision from J.L. and K.T. J.L., A.P.E., R.Z.C. and K.T. made all figures and wrote the manuscript.

## Competing interests

The authors declare no competing interests.
