## [Transparent Peer Review file · Nature Communications]

HSF-1 promotes longevity through ubiquilin-1 dependent mitochondrial network remodelling

Corresponding Author: Dr Johnathan Labbadia

Version 0:

Reviewer comments:

Reviewer #1

(Remarks to the Author)

The manuscript submitted by Erinjeri et al. is an interesting body of work surrounding HSF-1 and longevity in *C. elegans*. The investigators performed a genetic screen to determine modulators of HSF-1-dependent lifespan extension and found the proteasome shuttle factor Ubqln1 as an essential mediator which has no apparent effect on lifespan alone, but has a major effect on lifespan following HSF-1 overexpression. The authors go on to explore the putative mechanism behind this effect. They find that there is no major effect on protein aggregation but instead link ubqln1 to metabolic rewiring and posit that its role in mitochondrial dynamics governs the HSF-1 phenotype.

The methodology is sound and meets expected standards of rigor, and the work is of significance to the field as it brings new understanding to the interplay between proteostasis networks and metabolism. I think this is an interesting study that adds significantly to the UBQLN literature by building on an unexpected link between the two with implications for longevity. Clearly, the authors have done a considerable amount of detective work to understand how UBQLN1 influences HSF-1 dependent phenotypes. My significant critiques focus on an unmet need to clearly rule out a role for dysregulated protein degradation in the phenotypes observed, and a need to more directly link changes to mitochondrial homeostasis to UBQLN function and HSF-1 mediated lifespan extension.

I think these critiques can be answered with a few essential experiments, which would strengthen the manuscript's conclusions considerably.

Finally, I think it warrants acknowledgement that there is a hole in the manuscript – the identification of a protein or proteins which are accumulated upon UBQLN1 loss in a proteasomal-dependent fashion and contribute to the HSF-1 OE phenotype observed. This is mentioned in the Discussion, but regardless remains an unanswered question. However, given the repeated difficulty of investigators in finding clients of UBQLNs, it is not an insurmountable weakness to the manuscript and focusing on how specific pathways influence UBQLN-dependent longevity is a very valuable addition to the field.

Major points:

1. As the manuscript currently reads, the focus shifts to and from metabolism in Figures 3 and 5, which feels undermining to the researchers' focus on mitochondrial remodeling. Perhaps moving Figure 4 and 3 will help shift focus away from protein homeostasis to metabolism more cohesively.

2. I am unconvinced that metabolic rewiring is the cause for the reduced longevity of ubqln1 HSF-OE worms; rather, I find it equally likely that this is a downstream consequence of disrupted proteostasis which does not influence longevity, especially given the results from Figure 5d-f. Therefore, I find the conclusion from Figures 3-5 unsupported by the data presented. There either needs to be more experimental evidence for causation or the language needs to be changed.

I disagree that Figure 3 demonstrates that Ubqln1 is necessary for metabolic rewiring of HSF-1 OE worms. However, the way the proteomic data is presented in Figure 3e-f, it is hard for me to directly compare HSF-1 worms with and without UBQLN. In comparing hsfOE/hsfOE-ubqln1 in Table 2, there does not appear to be enrichment for changes in metabolic proteins. Instead, it appears that there are changes to integral membrane proteins with the largest change in npl-4, which would be consistent with a role for ubqln1 in ERAD.

Figure 5a shows the most compelling connection between UBQLN1, HSF-1, and mitochondrial rewiring. It would be particularly rewarding if there was a way to connect this OCR phenotype to other pathways, such as the Npl4 phenomenon in Figure 4b or the tunicamycin in Supplementary Figure 4b.

3. I am struck by what appears to be a network of protein folding and protein degradation apparatuses that modulate the effect of HSF-1 overexpression. Ubqln1 has the strongest modulatory effect on HSF-1 OE but in looking at the data in Supplementary Table 1, it is just one of a network. By comparing changes in survival of WT versus HSF-OE worms, Ubqln1 comes out at the top of a ranked list, closely followed by sgt-1, dnj-12, and sti-1. Ubqln1 interacts indirectly with SGTA in human cells to facilitate protein degradation (PMID: 27345149) and STI-1 shares homology with ubqln1 (which has 2 STI-1 domains) and also plays a role in protein degradation (reviewed in PMID: 32965492).

Along the same vein, I disagree with the conclusion that Ubqln1 is not necessary for enhanced proteostasis in Figure 2, which is discussed on page 4 lines 124-129. In fact, it appears that Figure 2f shows that Ubqln1 has some effect on polyQ35 accumulation in hsf-1 OE worms. In comparing Figure 2 and Supplementary Figure 2, it appears that there is an age-dependence of the aggregation phenotype. Is it possible that day 5 is too early to see Ubqln1-dependent changes to protein homeostasis?

For example, citation #25, Lim et al. 2009 show that Ubqln1 binds to erasin, which forms a complex with VCP to facilitate ERAD. They also use *C. elegans* as their model system and find in their Figure 9 that knockdown of Ubqln1 results in very high levels of polyubiquitinated protein in mixed-stage worms. It seems possible that the lack of a protein degradation phenotype in Figure 2 may be in part due to age or model system (polyQ reporter) selection.

In that manuscript, it is shown that UBQLN1 knockdown mimics tunicamycin exposure. This seems particularly relevant because of the finding here that HSF-1 OE worms are sensitive to tunicamycin compared to WT worms (Supplementary Figure 4b) and it may be worth mentioning.

A western blot of polyubiquitin in the tested worm strains would help to clarify this considerably. It would be useful in Figure 2 with the HSF-1 OE/UBQL-1 kd strains, in Figure 4 with the CDC48/NPL-4/UFD-1 strains, and in Supplementary Figure 4 with the induction of ER stress.

4. The focus on NHR-49 needs more rationale. On page 5 it is mentioned that UBQL-1 has been shown to interact strongly with NHR-49, but looking at this article's data it seems that UBQL-1 is ranked #900 of 4600 interactors of NHR-49. Does UBQL-1 bind to NHR-49 in worms? Does UBQL-1 loss lead to lower or higher levels of NHR-49 in worms by western blot? This possibility is hinted at on Page 5 line 166 but never examined. For me to be convinced that NHR-49 warrants particular focus, there needs to be a stronger experimental link.

5. Figure 4: are there changes to NPL-4 protein/CDC48/UFD1 in HSF and ubqln1 worms by western blot? The proteomics suggests that NPL-4 is up, but it is unclear how CDC48 and UFD1 are affected. It would be very powerful to have an orthogonal validation of the NPL-4 finding which also examines the other essential components of the system which are tested in Figure 4.

Minor points:

6. On page 3 line 75 there are some more comprehensive reviews of UBQLN function: PMCID: PMC7737201 and PMID: 22628307.
7. It is also worth mentioning that STI1 domains of UBQLNs help govern client binding, and that it is not simply binding of ubiquitinated protein on page 3 line 78.
8. There may be a missed opportunity in Figure 1 to discuss genes that have a wt lifespan phenotype which is mitigated by HSF-1 OE. Are they in the same pathway as UBQLN1 for example?
9. On page 6 line 189: the data shown have not proven an impairment of organellar protein degradation. The best way to show that would be with a pulse-chase, but that can require considerable troubleshooting and might be a major undertaking. The language just needs re-wording.
10. Figure 3: the x-axis label for a and c needs to be more descriptive. Is this log2 ratio?
11. Figure 3: I would change the order to go from genes > proteins. Part e/f should go first.
12. Figure 3: what are the scales for d/f? Is that a z score?
13. Figure 3: the graphical representation of part e is hard to understand. Why not show another volcano plot here?
14. Figure 5 b-c don't all add up to 100? The top line of some bars is missing.
15. As it stands, Figure 5d-f seems more appropriate for a Supplement since the connection is unclear and the data are mostly negative.
16. Supplementary Figure 1c needs to show STI1 domains of UBQL-1.
17. In Supplementary Figure 3c there is some confusion about whether the graph is showing UBQLN-1 dependence of HSF-1 phenotype or simply KEGG pathways changed by HSF-1
18. The title of Supplementary Figure 5 is too long and not an adequate summary of the results.
19. On Page 8 it is mentioned that it's surprising that they did not identify a greater number of proteins with altered abundance (lines 247-249). However, this could be because the proteomics done here were not with isolated cytosol as mentioned on line 247.
20. The discussion mentions a possible interaction between UBQLN1 and the cytoskeleton. In fact, mammalian UBQLNs

were first identified as an interactor of vimentin (PMID: 10549293).

Reviewer #2

(Remarks to the Author)

Annamary Paul Erinjeri and co-workers describes how HSF-1 promotes lifespan in *C. elegans*. Using an RNAi screens, they have identified ubiquilin-1 as a downstream factor of hsf-1. They observed that hsf-1 overexpression leads to transcriptional downregulation of all components of the CDC-48-UFD-1-NPL-4 complex, which is important to endoplasmic reticulum and mitochondria associated protein degradation.

The finding that enhanced HSF1 expression increases lifespan is quite interesting but a mechanism how the HSF1 protein regulate ubiquitin expression and how ubiquitin acts (directly) on further downstream proteins remains unclear. In addition, the authors switch between altered protein/gene expression levels and "activity" which is sometimes difficult to follow. One has to be careful to use the term "activity" since the authors have not tested whether genes/proteins of interest have a higher activity based on PTMs, interactions, translocation or alternative activity assays.

Why have the authors mentioned the NHR-49 as an interactor of ubiquitin? Is this factor regulated in the transcriptomics or proteomics dataset? Later the authors mentioned an affected lipid metabolism. Do the authors detect also other genes/proteins associated to fatty acid metabolism in their gene/protein expression data? It might be worse to select one affected candidate (ACS2?) and test for protein ubiquitination. Have the authors tested a general change in protein ubiquitination in their worm models? either by immunoblotting or a diglycine remnant IP combined with LC-MS? In case hsf1 has a direct effect on mitochondrial proteins, is HSF1 localized in mitochondria? Or do the authors observe changes in protein ubiquitination in mitochondria? The experiments with the protein aggregation are interesting but how correlate this finding with the remodelling of mitochondria? Is it possible to overlap the aggregate staining with the mitochondrial and or ER-network?

The authors observed a reduced level of ACS2 and postulate that an altered activity of NHR49 might be responsible.

However, this should be verified by some experiments. Do the author observed a general down regulation of mitochondria in hsf-1 OE and double mutant worms? This could be tested by the levels of mtDNA and labelling all mitochondrial proteins in the volcano plot. A box plot might reveal whether these changes of mitochondrial proteins is significant (compare whole protein distribution versus all detected mitochondrial proteins in a box plot and calculate significances).

Since the mitochondrial remodelling is dramatically changed and the extensive remodelling of mitochondria is mentioned in the abstract, why have the authors hid this data in the supplement? Some immunostainings with a higher magnification might be helpful here to better visualize the changes of mitochondria.

Minor:

The transcriptomics and proteomics analysis the data are not well documented in the SI material. The description of the axis of Figure 2 are not proper described. For example, one should use $-\log_{10}$ p-values and what means "difference? What is the bended line within the volcano plot? Please explain briefly at least in the method part the FDR calculation and how the bended line was generated. A more systematic analysis such as a PCA or unsupervised hierarchical clustering might be helpful to judge the significance of the observed changes.

The header/identifier of table 3 is not proper described. What means "#Name?", difference, and what values are in column L-S? The table should contain all proteins which were quantified, independent of their ratio and p-value. This might help to judge the depth of the proteome and shows which factors are not affected. This is mandatory for the supplement.

The colour code of protein changes in Figure 3 is not explained. What means -5 in Figure 3d?

Page 5 line 144-148: The authors report that loss of ubiquitin results in 43 increased and 25 decreased proteins. Then they described the RNA-Seq data as "Similarly", which shows 10x more upregulated genes?

The authors stated "we first investigated whether ubiquilin-1 influences HSF-1 activity. Loss of ubql-1 function did not suppress the expression of canonical HSF-1 target genes... demonstrating that HSF-1 activity is unaffected by the loss of ubql1 activity". It is highly speculative to judge the activity by the expression level of two target proteins. HSF1 might affect also other proteins. Have the authors identified some PTMs on HSF1 or do the authors observed an altered ubiquitination of HSF1 or other proteins? Do the authors detect a general change in protein ubiquitination by immunoblotting of cellular extracts or isolated organelles?

Do the authors observe a change in lipid metabolism in hsf1 OE worms? It might useful to verify changes in lipid synthesis by a lipidomics/metabolomics analysis.

Reviewer #3

(Remarks to the Author)

In this manuscript, the authors investigate how overexpression of heat shock factor, HSF-1, extends lifespan in *C. elegans*. They started with a RNAi screen and identified that ubql-1 is required for lifespan extensions of hsf-1 OE animals. By transcriptomics and proteomics analyses, they claimed that genes and proteins in metabolic pathway are upregulated in hsf-1 OE animals, which is dependent on ubql-1 and its potential action on nhr-49. Since UBQL-1 is involved in protein degradation they focused on proteins whose reduction in hsf-1 OE animals was dependent on ubql-1, namely NPL-4.1/2. To further characterized the role of CDC-48-NPL-4-UFD in HSF-1 OE animals. Then, they turned their focus on metabolic and mitochondrial (dynamics) changes. ubql-1 mutation in hsf-1 OE worms restored OCR and ATP level to wild type and reduced the level of lipid droplets (measured by Oil-Red-O staining). The loss of ubql-1 also altered muscular mitochondrial morphologies. The overexpression of HSF-1 induced an increase in the number of elongated/tubular mitochondria, while

ubql-1 mutation compromised this effect. RNAi against fission not fusion genes inhibited the lifespan extension in hsf-1 OE worms and knockdown of mitophagy did not change the lifespan of hsf-1 OE.

While this study demonstrates a role for HSF-1 OE in mitochondrial remodeling, the role of ubql-1 in this process remains unclear. Further it is unclear how reduced mitochondrial fission suppresses hsf-1 OE lifespan extension.

1. Ubiquitin-1 is required for HSF-1 mediated lifespan extension

- The RNAi screen is not described in enough detail and missing from the methods section. How much FUdR did the authors add? How did they evaluate the outcomes of RNAi? How did they measure the lifespan on 12-wells plates?
- Since ubql-1 is an HSF-1 target gene, they authors should show that ubql-1 transcription is mediated by hsf-1. Is ubql-1 transcription reduced upon hsf-1 RNAi or in hsf-1 mutants?
- Where is ubql-1 expressed, this is important since the authors are determining tissue-specific proteostasis models and mitochondrial measures.
- Is ubql-1(tm1574) a null mutant? The authors showed this mutant has truncated mRNA. Does this mutant have truncated or no UBQL-1 protein?
- Is ubql-1 activity increased in HSF-1 OE animals? While the authors are trying to get to this with their proteomics analysis, it's in a rather confusing and roundabout way.

2. Maintenance of proteostasis capacity does not require ubql-1 function

- Line 105-110: Since ubql-1 is regulated by HSF-1, why do the authors think loss of ubql-1 would affect HSF-1 activity? Is there any feedback effect? Does any literature support this?
- The data in Figure 2a shows that ubql-1 mutants show an intermediate phenotype between WT and HSF-1 OE in hsp-16.11 and hsp-70 levels. What is the conclusion here? That the expression is equal to WT or equal to HSF-1 OE? This data needs to be strengthened.
- Line 115-117, "Given that ubql-1 mutants do not exhibit increased HSF-1 activity, these data suggest that ubql-1 is necessary for increased stress resistance in hsf-1 OE worms" This statement is not true. Mutation of ubql-1 in hsf-1 OE animals did not change the survival after heat shock or the level of polyQ aggregates (fig 2c-2f). However, it did shorten the survival in wild type, suggesting that ubql-1 is required for heat stress survival in basal condition (fig 2c).
- Fig 2d-f, although the ubql-1(RNAi) does not affect polyQ aggregation in neither wild type nor hsf-1 OE animals in intestine, it did affect PolyQ aggregation in body wall muscle (at least in one experiment, how many experiments were conducted in total? More repeats may be necessary), did the authors try ubql-1 mutant? Based on their lifespan analysis, the ubql-1 mutant appears to show stronger phenotype comparing the lifespan of ubql-1 RNAi in fig 1c-d to ctrl and that of ubql-1 mutant with wild type in fig 4b-d.

3. Ubiquitin-1 promotes metabolic remodelling in hsf-1 OE animals

- Is it necessary to include the transcriptomic analysis? What is the rationale for transcriptomic analyses? Given that ubql-1 is important for protein degradation, it seems that the proteomics should drive the manuscript.
- Why did the authors use different -logP in Fig 3a and 3c? Different criteria or statistics?
- What are the Gene Ontology (GO) analyses of genes and proteins that upregulated in hsf-1 OE but downregulated in hsf-1 OE; ubql-1(-) and vice versa? By examining this, we can learn what ubql-1 regulates in hsf-1 OE worms.
- What happened to a further characterization and follow up of NHR-49? The authors pose the hypothesis that ubql-1 could alter the stability of NHR-49 complexes, and spend some time in the discussion to integrate this into their data, but additional experiments linking NHR-49 activity, HSF-1 OE, ubql-1 and the mitochondrial phenotypes are warranted.

4. Ubiquitin-1 promotes down-tuning of endoplasmic reticulum and mitochondrial associated degradation components in hsf-1 OE animals

- Line 173: "Therefore, we focused our attention on proteins whose levels were elevated in hsf-1 OE animals upon reduction of UBQL-1 function. Among the proteins whose levels are decreased upon hsf-1 OE, we identified 4 whose reduction was dependent on UBQL-1." This description is confusing. The authors should consider rephrasing: to something like: Since we hypothesize that UBQL-1 activity is increased in HSF-1 OE animals, which would lead to increased degradation of UBQL-1 target proteins, we focused our attention on proteins whose levels were decreased in hsf-1 OE animals in a manner dependent on UBQL-1 function."
- The authors conclusion that HSF-1 OE worms are long-lived due to "mild impairment of organellar protein degradation" is not convincing. Can the authors show that "mild impairment of organellar protein degradation" can extend lifespan?
- The authors need to explain the function of the CDC-48/NPL-4/UFD complex better in both the context of ERAD and MAD (and show that this function is changed upon NPL-4 reduction with and without HSF-1 OE)
- The authors are hypothesizing that HSF-1 OE animals have increased UBQL-1 function which leads to increased degradation of NPL-4., which means that HSF-1 OE, ubql-1 mutants should have restored longevity after depletion of NPL-4. The authors however show a further decrease in lifespan. While it is possible that this reviewer is missing something here, in any case this section needs to be much better explained.

5. HSF-1 overexpression promotes lifespan by altering mitochondrial dynamics

- The quality of mitochondrial images in sup Fig 4c and Fig 5d should be improved, since the authors largely determined whether ubql-1 mutation induces mitochondrial fragmentation in hsf-1 OE animals or not based on the images. Using confocal microscopy should get a higher resolution image.
- How many repeats of mitochondrial morphology experiments were done? No error bars were shown in quantification results and the authors didn't mention how many repeats in either legends nor methods.
- Why did they focus on the region between the pharynx and vulva or the vulva and tail for muscular mitochondria quantification? Because of these regions show vigorous mitochondrial activity? Or is it just easier to image comparing with

other regions? They use body wall muscle promoter (myo-3) to drive mitochondrial GFP and should observe GFP all over the whole body in muscle.

6. Rigor:

- Some of the experiments (e.g. fig 2e-f, sup fig 2c-e, sup fig 4e, and sup fig 5b & c) were only done by two repeats. We strongly recommend the authors to add at least one more repeat.
- The quantification results (bar charts) in fig 5b and 5c are exactly the same in sup fig 5e and 5f?

7. Minor comments

- What is the lifespan after knocking down mitochondrial fission/fusion genes in hsf-1 OE; ubql-1(tm1574)?
- Does UBQL-1 affect mitochondrial dynamics by regulating fission/fusion genes?
- Fig 2d, we encourage the authors to add ubql-1(RNAi) in figure to help reader better understand it is RNAi not mutant even the authors mentioned in legends.
- Fig 2 e and 2f, the authors can put full text of intestine and body wall muscle in figures, not just using initial i or m to present it.
- Fig 3, we encourage that the authors put the names of those proteins that are highly differentiated in volcano plots.

Reviewer #4

(Remarks to the Author)

Version 1:

Reviewer comments:

Reviewer #1

(Remarks to the Author)

The authors of "HSF-1 promotes longevity through ubiquitin-1 dependent mitochondrial network remodeling" have gone through extensive revision of their manuscript with considerable editorial changes and experimental additions. The manuscript only needs minor editorial revision.

Regarding the original major points, all of my concerns have been addressed. In particular, the addition of polyubiquitin blots enriches the study by adding an understanding of how different forms of protein degradation (the proteasome with K48 blots and autophagy with K63 blots) are influenced by hsf-1 and ubql-1.

Regarding minor point 10, Figure 3 could use minor additional clarification on which direction the log2 ratios are measuring in (a) and (b), for ease of interpretation. If the log2 ratio is hsf-1 OE;ubql1(tm1574) / hsf-1 OE then it would be helpful to include that, as it clarifies that genes/proteins on the positive side are accumulated upon ubql1 perturbation.

On page 8 Figure 3f and 3e are discussed/introduced in non-chronological order.

Reviewer #2

(Remarks to the Author)

Although the authors have still not found a direct mechanism, all of the reviewer's questions have been sufficiently answered

Reviewer #3

(Remarks to the Author)

The authors have made a commendable effort in addressing the reviewers' comments, resulting in a substantially improved manuscript. Overstatements have been appropriately removed. While they added some new experiments, they largely sidestepped the more extensive experimental work suggested by the reviewers, justifying this approach in their responses. Nonetheless, the revisions have strengthened the manuscript overall, and I recommend it for publication.

Reviewer #4

(Remarks to the Author)

Point-by point responses to reviewer comments

REVIEWER COMMENTS

Reviewer #1 (Remarks to the Author):

The manuscript submitted by Erinjeri et al. is an interesting body of work surrounding HSF-1 and longevity in *C. elegans*. The investigators performed a genetic screen to determine modulators of HSF-1-dependent lifespan extension and found the proteasome shuttle factor Ubqln1 as an essential mediator which has no apparent effect on lifespan alone but has a major effect on lifespan following HSF-1 overexpression. The authors go on to explore the putative mechanism behind this effect. They find that there is no major effect on protein aggregation but instead link ubqln1 to metabolic rewiring and posit that its role in mitochondrial dynamics governs the HSF-1 phenotype.

The methodology is sound and meets expected standards of rigor, and the work is of significance to the field as it brings new understanding to the interplay between proteostasis networks and metabolism. I think this is an interesting study that adds significantly to the UBQLN literature by building on an unexpected link between the two with implications for longevity. Clearly, the authors have done a considerable amount of detective work to understand how UBQLN1 influences HSF-1 dependent phenotypes. My significant critiques focus on an unmet need to clearly rule out a role for dysregulated protein degradation in the phenotypes observed, and a need to more directly link changes to mitochondrial homeostasis to UBQLN function and HSF-1 mediated lifespan extension. I think these critiques can be answered with a few essential experiments, which would strengthen the manuscript's conclusions considerably.

Finally, I think it warrants acknowledgement that there is a hole in the manuscript – the identification of a protein or proteins which are accumulated upon UBQLN1 loss in a proteasomal-dependent fashion and contribute to the HSF-1 OE phenotype observed. This is mentioned in the Discussion, but regardless remains an unanswered question. However, given the repeated difficulty of investigators in finding clients of UBQLNs, it is not an insurmountable weakness to the manuscript and focusing on how specific pathways influence UBQLN-dependent longevity is a very valuable addition to the field.

We thank the reviewer for taking the time to assess our work and for providing genuinely thoughtful, reasonable, insightful and helpful feedback. The criticisms raised were all fair, and we have done our best to address them as detailed below. We think that our manuscript is greatly improved as a result of your comments.

Major points:

1. As the manuscript currently reads, the focus shifts to and from metabolism in Figures 3 and 5, which feels undermining to the researchers' focus on mitochondrial remodeling. Perhaps moving Figure 4 and 3 will help shift focus away from protein homeostasis to metabolism more cohesively. Thank you for pointing this out. When reading back through the manuscript it was evident that the flow of our work was quite clunky. Therefore, we have taken your suggestion and moved data from Figures 3/S3, 4/S4 and 5/S5 around, to make the transition from proteostasis, to NPL-4.1 levels, changes in mitochondrial dynamics and longevity, smoother. Specifically, we have used Figure 3 to show that *hsf-1* OE reduces NPL-4.1 levels in a *ubql-1*-dependent manner, Figure 4 to show that this is associated with increased mitochondrial fusion in early adulthood, and Figure 5 to show that the increased mitochondrial fusion and lifespan of *hsf-1* OE worms are dependent on *ubql-1*.

2. I am unconvinced that metabolic rewiring is the cause for the reduced longevity of ubqln1 HSF-OE worms; rather, I find it equally likely that this is a downstream consequence of disrupted proteostasis which does not influence longevity, especially given the results from Figure 5d-f. Therefore, I find the conclusion from Figures 3-5 unsupported by the data presented. There either needs to be more experimental evidence for causation or the language needs to be changed.

We agree that this is an overstatement and needed to be softened. Therefore, we have removed this statement from the text and replaced it with more accurate conclusions within the results and discussion sections as follows:

Results (page 7, line 13): *“These results indicate that ubql-1 regulates the expression/abundance of genes/proteins associated with diverse processes, with membrane associated and fatty-acid biosynthesis proteins affected.”*

Discussion (page 12, line 11): *“Future work to understand how, if at all, changes in ACS-2 and other ubql-1 dependent metabolic processes impact hsf-1 OE lifespan, will reveal whether these changes are causal to lifespan extension.”*

We have also changed sub-headings and figure legend titles as follows:

(sub-heading, page 6, line 9): *“Ubql-1 is necessary for a sub-set of transcriptomic and proteomic changes induced by hsf-1 overexpression”*

Legend title for Figure 3: “Ubql-1 is necessary for elevated NPL-4.1 levels in hsf-1 OE animals”

Legend title for Supplementary Figure 3: “hsf-1 overexpression leads to changes in genes and proteins associated with metabolic homeostasis”

I disagree that Figure 3 demonstrates that Ubqln1 is necessary for metabolic rewiring of HSF-1 OE worms. However, the way the proteomic data is presented in Figure 3e-f, it is hard for me to directly compare HSF-1 worms with and without UBQLN. In comparing hsfOE/hsfOE-ubqln1 in Table 2, there does not appear to be enrichment for changes in metabolic proteins. Instead, it appears that there are changes to integral membrane proteins with the largest change in npl-4, which would be consistent with a role for ubqln1 in ERAD.

We apologise for not making the direct comparisons between *hsf-1* OE and *hsf-1* OE; *ubql-1* mutant worms as straightforward as it should have been. We also noticed that one of our panel titles was incorrect and therefore confusing. To rectify this, we have reworked the volcano plots in Figures 3 and S3 to label the genes/proteins showing the largest changes in expression/abundance between the *hsf-1* OE and *hsf-1* OE; *ubql-1* strains. We have also rearranged the positions of figure panels to allow for a more streamlined and direct comparison of data from the *hsf-1* OE and *hsf-1* OE; *ubql-1* mutant strains (these are now presented side by side in new Figure 3a and b).

We also agree that loss of *ubql-1* only changes a small number of metabolic genes/proteins in *hsf-1* OE worms. KEGG analysis for pathway enrichment was impossible on *ubql-1*-dependent genes/proteins as the numbers were too small. However, we were able to run gene ontology and WormFlux analysis (which looks for enrichment in *C. elegans* metabolic pathways) on our data. This revealed the terms “Cell projectile organization”, “actin filaments”, “defense response”, “nicotine and nicotinamide metabolism” and “fatty acid biosynthesis”

among the genes and proteins altered in *hsf-1* OE worms upon loss of *ubql-1*. This information can be found in new Supplementary Tables 4 and 7.

Furthermore, we have taken your lead/advice and point out that 8 of the proteins that exhibit altered abundance in *hsf-1* OE; *ubql-1* mutants are integral membrane proteins. This new information is now provided in the results section as follows:

Page 7, Line 1: *“Gene ontology analysis revealed that ubql-1 dependent genes in hsf-1 OE worms were enriched for the terms “cell projection organization” (p < 0.00018) and “defense response” (p<0.00022), while differentially regulated proteins were enriched for the term “actin-based filaments” (Supplementary Table 4 and Supplementary Table 7). The enrichment for “metabolic pathways” among hsf-1 OE mediated changes in gene expression and protein abundance prompted us to use WormFlux²⁵ to investigate whether any metabolic pathways were significantly altered by loss of ubql-1. This revealed “Fatty acid biosynthesis” (ACS-2, PPT-1; P-enrichment = 0.00096) and “nicotine and nicotinamide metabolism” (parg-2; P-enrichment = 0.038) as pathways that are disrupted in hsf-1 OE worms lacking ubql-1 (Supplementary Table 4 and Supplementary Table 7). Finally, an assessment of sub-cellular localization revealed that of the 43 proteins with altered abundance in hsf-1 OE; ubql-1 mutants, 8 were annotated as “integral membrane proteins” (Supplementary Table 3). These results indicate that ubql-1 regulates the expression/abundance of genes/proteins associated with diverse processes, with membrane associated and fatty-acid biosynthesis proteins affected.”*

Figure 5a shows the most compelling connection between UBQLN1, HSF-1, and mitochondrial rewiring. It would be particularly rewarding if there was a way to connect this OCR phenotype to other pathways, such as the *Npl4* phenomenon in Figure 4b or the tunicamycin in Supplementary Figure 4b.

We agree. To address this, we performed OCR and mitochondrial morphology experiments using *npl-4.1/4.2* RNAi. We find that knockdown of *npl-4.1/4.2* is sufficient to increase mitochondrial fusion in wild type animals but does not further increase mitochondrial fusion in *hsf-1* OE animals. This supports our model, whereby *hsf-1* OE leads to reduced NPL-4.1 levels, which promotes mitochondrial fusion. However, *npl-4.1* RNAi did not have a concomitant effect on OCR. This raises two possibilities – either OCR changes at later time points when knocking down *npl-4.1* in wild-type worms, or increased mitochondrial fusion is not sufficient to provoke changes in OCR in wildtype worms. These data are included as new Figure 5e and f.

3. I am struck by what appears to be a network of protein folding and protein degradation apparatuses that modulate the effect of HSF-1 overexpression. Ubqln1 has the strongest modulatory effect on HSF-1 OE but in looking at the data in Supplementary Table 1, it is just one of a network. By comparing changes in survival of WT versus HSF-OE worms, Ubqln1 comes out at the top of a ranked list, closely followed by *sgt-1*, *dnj-12*, and *sti-1*. Ubqln1 interacts indirectly with SGTA in human cells to facilitate protein degradation (PMID: 27345149) and STI-1 shares homology with ubqln1 (which has 2 STI-1 domains) and also plays a role in protein degradation (reviewed in PMID: 32965492).

This is a fantastic point - we are very grateful for the interesting additional information and for highlighting this. To make use of this, we used STRING to create an interaction network of all RNAi hits that reduced *hsf-1* OE lifespan by at least 20% without a corresponding decrease in wildtype lifespan. We focused on interactions that were based on evidence of co-expression and/or experimental interaction. Any hits that did not show any connections under these criteria were excluded from the network. These new data are now included in new Figure 1b.

In response to your minor comment 8, we also generated an interaction network for RNAi clones that increase lifespan *hsf-1* OE worms but reduce lifespan in wildtype worms (with *ubql-1* included to look for potential interactors in this second set of genes). This revealed a small set of genes that interact with *ubql-1*, including the ribosome associating factor MBF-1/EDF1 and the ribosomal subunit RPS-1/RPS3A. We have included this additional network in new Figure 1c.

We also tried to generate a network for RNAi clones that increased lifespan in wild type worms but reduced or did not alter *hsf-1* OE lifespan. However, of the three hits in this category, we found no evidence of any interaction with *ubql-1* or each other, and therefore did not include this in the paper.

We have modified the results section as follows to reflect these new data:

Page 3, line 32: *“An analysis of all RNAi clones that reduced the lifespan of hsf-1 OE worms > 20% but did not alter wild-type lifespan, revealed a network (based on co-expression or physical interaction) formed of two clusters: one containing UBQL-1 and SGT-1, and another containing STI-1, DNJ-12, CCT-2, FKB-6 and ABCF-2 (Fig. 1b and Supplementary Table 1). Conversely, we also identified genes whose knockdown increased hsf-1 OE lifespan > 20% but had an opposing effect on the lifespan of wildtype worms. These were found to contain a small ubiquitin-associated network (co-expression or physical interaction) containing the ribosome associated proteins RPS-1 and MBF-1¹⁷, as well as the co-chaperone DAF-41/P23 and the protein disulfide isomerase, PDI-3 (Fig. 1c and Supplementary Table 1).*

UBQLN1 has been shown to directly interact with SGTA to mediate protein degradation in human cells¹⁸, while STI1 has homology with UBQLN1 and has roles in protein degradation¹². Together, these findings suggest that UBQL-1 may act as part of, or in parallel to, a wider network of protein folding and degradation complexes to promote longevity upon hsf-1 OE.”

Along the same vein, I disagree with the conclusion that Ubqln1 is not necessary for enhanced proteostasis in Figure 2, which is discussed on page 4 lines 124-129. In fact, it appears that Figure 2f shows that Ubqln1 has some effect on polyQ35 accumulation in *hsf-1* OE worms. In comparing Figure 2 and Supplementary Figure 2, it appears that there is an age-dependence of the aggregation phenotype. Is it possible that day 5 is too early to see Ubqln1-dependent changes to protein homeostasis? For example, citation #25, Lim et al. 2009 show that Ubqln1 binds to erasin, which forms a complex with VCP to facilitate ERAD. They also use *C. elegans* as their model system and find in their Figure 9 that knockdown of Ubqln1 results in very high levels of polyubiquitinated protein in mixed-stage worms. It seems possible that the lack of a protein degradation phenotype in Figure 2 may be in part due to age or model system (polyQ

reporter) selection. In that manuscript, it is shown that UBQLN1 knockdown mimics tunicamycin exposure. This seems particularly relevant because of the finding here that HSF-1 OE worms are sensitive to tunicamycin compared to WT worms (Supplementary Figure 4b) and it may be worth mentioning. A western blot of polyubiquitin in the tested worm strains would help to clarify this considerably. It would be useful in Figure 2 with the HSF-1 OE/UBQL-1 kd strains, in Figure 4 with the CDC48/NPL-4/UFD-1 strains, and in Supplementary Figure 4 with the induction of ER stress.

With regards to our conclusions related to the role of *ubql-1* in the suppression of polyQ aggregation by *hsf-1* OE, we agree that our language was too imprecise. In absolute terms, it is true that *ubql-1* (RNAi) leads to a modest increase in polyQ aggregation in *hsf-1* OE muscle tissues on day 3 and day 5 of adulthood. Our point is that by day 5 of adulthood, the ability of *hsf-1* OE to suppress aggregation is unaffected (i.e. the reduction in aggregates between WT and *hsf-1* OE worms is the same when comparing the control and *ubql-1* RNAi groups. We have now modified the sub-heading and text within the corresponding results section as follows to more precisely describe our data:

Sub-heading, page 4, line 30: "*Ubql-1* is not required for *hsf-1* OE to suppress age-related protein aggregation"

Results, Page 5, line 14: "*PolyQ* aggregation increased with age in both intestinal and muscle tissues (Fig. 2e-g and Supplementary Fig. 2c-e) of wildtype worms and was strongly suppressed on day 3 and day 5 of adulthood by overexpression of *hsf-1* (Fig. 2e-g and Supplementary Fig. 2c-e). Knockdown of *ubql-1* increased polyQ aggregation on day 5 of adulthood within the muscle, but not the intestine, of both wild type and *hsf-1* OE worms (Fig. 2e-g and Supplementary Fig. 2c). However, the degree to which *hsf-1* OE suppressed polyQ aggregation in muscle and intestinal tissues compared to wildtype counterparts, was the same in both the control (RNAi) and *ubql-1* (RNAi) treatment groups (Fig. 2f and g and Supplementary Fig. 2d and e). This suggests that *ubql-1* is not necessary for *hsf-1* OE to suppress age-related polyglutamine aggregation."

We also agree that we placed too much emphasis on data from a single proteostasis sensor (polyQ::YFP) that may or may not reflect what is happening to polyubiquitylated proteins in ageing animals. To address this, we have now assessed levels of total polyubiquitin, K48-linked ubiquitin (proteasome targeted) and K63-linked ubiquitin (lysosome targeted) in *hsf-1* OE and *hsf-1* OE; *ubql-1* mutant worms on day 1 and day 5 of adulthood, using SDS-PAGE and western blotting.

These experiments revealed that aggregated polyUb, K48-linked Ub and K63-linked Ub all increase between day 1 and day 5 of adulthood. Furthermore, while *ubql-1* mutation or *hsf-1* OE did not consistently alter levels of soluble or aggregated polyUb or K63-linked Ub proteins on day 1 or day 5 of adulthood, loss of *ubql-1* lead to an increase in K48-linked ubiquitylated proteins on day 1 and day 5 of adulthood. Interestingly, *hsf-1* OE suppressed K48-linked Ub protein aggregation on day 5 of adulthood but did so even in the *ubql-1* mutant background (although loss of *ubql-1* does appear to increase absolute levels of K48-linked Ub aggregates in *hsf-1* OE worms on day 5 of adulthood). These data are consistent with our muscle polyQ experiments. In addition, given that the loss of *ubql-1* leads to an increase in K48-linked ubiquitylated proteins on day 1 of adulthood in wildtype animals, we do not think that the

timepoints we have chosen are too early. These new data are included in new Fig. 2h and i and Supplementary Fig. 2f-m). We have updated the results text to reflect these new data as follows:

*Page 5, line 25: "Given that polyQ is an artificial sensor of proteostasis capacity, we reasoned that loss of *ubql-1* may impede the ability of *hsf-1* OE to promote the degradation of endogenous ubiquitylated substrates with age. Therefore, we assessed the abundance of polyubiquitylated, K48-linked ubiquitylated (proteasome directed) and K63-linked ubiquitylated (lysosome directed) proteins²³, on day 1 and day 5 of adulthood.*

*We observed that levels of high molecular weight, aggregated polyUb, K48-ubiquitylated and K63-ubiquitylated proteins increased with age in all genotypes tested (Fig. 2h, i and Supplementary Fig. 2f-m). However, the level of soluble and aggregated polyubiquitylated or K63-ubiquitylated proteins was not consistently altered on day 1 or day 5 of adulthood in *hsf-1* OE or *ubql-1* mutant animals, compared to wild type controls (Supplementary Fig. 2f-m).*

*In contrast, the levels of both soluble and aggregated K48-ubiquitylated proteins was elevated in *ubql-1* mutant worms on day 1 and day 5 of adulthood (Fig. 2h, i and Supplementary Fig. h, i). Interestingly, *hsf-1* OE did not consistently alter levels of K48-Ub proteins on day 1 of adulthood, but did reduce levels of both soluble and aggregated K48-Ub proteins in both wild type and *ubql-1* mutant backgrounds on day 5 of adulthood (Fig. 2h, i and Supplementary Fig. 2h, i). These data suggest that both *ubql-1* and *hsf-1* OE suppress the accumulation and aggregation of K48-linked ubiquitylated proteins with age, but that *ubql-1* is not necessary for *hsf-1* OE to mediate these effects."*

Lastly, a direct comparison of our findings with those reported in Lim et al. 2009 is impossible as they performed experiments in mixed stage GFP-reporter animals after long-term exposure to *ubql-1* RNAi. This is a completely different scenario than in our experiments, where worms were age-synchronised and exposed to *ubql-1* RNAi for a shorter period of time, or to a *ubql-1* mutation that has been present throughout life.

4. The focus on NHR-49 needs more rationale. On page 5 it is mentioned that UBQL-1 has been shown to interact strongly with NHR-49, but looking at this article's data it seems that UBQL-1 is ranked #900 of 4600 interactors of NHR-49. Does UBQL-1 bind to NHR-49 in worms? Does UBQL-1 loss lead to lower or higher levels of NHR-49 in worms by western blot? This possibility is hinted at on Page 5 line 166 but never examined. For me to be convinced that NHR-49 warrants particular focus, there needs to be a stronger experimental link.

We completely agree with this point. Our attention was directed towards NHR-49 for two reasons: (i) recently published work demonstrated that *nhr-49* is necessary for the increased lifespan of *hsf-1* OE worms (PMID: 36261024) and increased HSF-1 activity in germline deficient worms (PMID: 37162952), and (ii) we observed that *ubql-1* is necessary for increased ACS-2 levels (a key mitochondrial membrane protein and canonical marker for NHR-49

activity) in *hsf-1* OE worms. Therefore, we assumed that our findings fed into previous observations; interestingly, this assumption appears to be wrong.

We crossed a well-established transcriptional reporter for NHR-49 activity (*acs-2* promoter driving GFP expression) into our *hsf-1* OE background, and exposed animals to *ubql-1* RNAi. As expected, *nhr-49* RNAi suppressed reporter activity in both wildtype and *hsf-1* OE backgrounds. Surprisingly, we also found that *hsf-1* OE reduced reporter activity; however, this was unaltered by the presence of *ubql-1* (RNAi). These results suggest an interesting model where loss of *ubql-1* leads to reduced ACS-2 levels independently of regulating NHR-49 activity. These new data are presented as new Supplementary Fig. 3i and j.

We have also modified the corresponding results and discussion sections as follows to reflect this:

Results, page 7, line 26: *“Acs-2 expression is strongly controlled by the transcription factor NHR-49, which has been shown to be necessary for lipid homeostasis and increased lifespan, including in hsf-1 OE worms^{9,28-33}. Therefore, we hypothesized that ubql-1 may be exerting a positive effect on lifespan by promoting NHR-49 activity.*

As expected, we found that nhr-49 (RNAi) shortens the lifespan of wildtype worms and prevents hsf-1 OE-mediated lifespan extension (Supplementary Fig. 3g and h and Supplementary Table 2). However, knockdown of ubql-1 did not alter NHR-49 activity in hsf-1 OE worms (Supplementary Fig. 3i and j), suggesting that ubql-1 does not influence ACS-2 levels or lifespan in hsf-1 OE animals through regulation of NHR-49 activity.”

Discussion, page 12, line 1: *“We also identified several proteins whose abundance decreased in hsf-1 OE worms upon loss of ubql-1. Among the most reduced proteins was the mitochondrial acetyl CoA-synthetase, ACS-2, which has a core role in fatty-acid metabolism and whose expression is controlled by the transcription factor, NHR-49^{9,53}. Given that nhr-49 and altered lipid metabolism are necessary for the increased lifespan of C. elegans possessing enhanced HSF-1 activity^{51,52}, it is interesting that we did not see any effect of ubql-1 RNAi on NHR-49 activity. However, we did find that ubql-1 was necessary for normal fat levels and elevated ACS-2 levels (a key target of NHR-49) in hsf-1 OE worms. ACS-2 plays an important role in regulating fatty-acid metabolism but is not required for normal lifespan⁵³. However, ACS-2 does mediate lifespan extension in response to lysosomal signalling in C. elegans⁵³. Future work to understand how, if at all, changes in ACS-2 and other ubql-1 dependent metabolic processes impact hsf-1 OE lifespan, will reveal whether these changes are causal to lifespan extension.”*

5. Figure 4: are there changes to NPL-4 protein/CDC48/UFD1 in HSF and ubqln1 worms by western blot? The proteomics suggests that NPL-4 is up, but it is unclear how CDC48 and UFD1 are affected. It would be very powerful to have an orthogonal validation of the NPL-4 finding which also examines the other essential components of the system which are tested in Figure 4.

This is another very good point. To address this, we first returned to our proteomics data and pulled out the abundance values for NPL-4.1/2, CDC-48.1/2, and UFD-1/2 in *hsf-1* OE and *hsf-1* OE; *ubql-1* mutant animals. We did not see any evidence for a change in the abundance of any of these proteins except NPL-4.1 in *hsf-1* OE; *ubql-1* mutant worms. Therefore, we did not see any reason to pursue western blotting for CDC-48 or UFD-1. These new data are now included as new Figure 3f.

Frustratingly, we are not aware of any in-house or commercially available antibodies raised against *C. elegans* NPL-4.1. As a potential alternative, we looked for commercially available antibodies that had been raised against well-conserved epitopes within human NPLOC4. We found one potential antibody that we tested in wildtype animals exposed to *npl-4.1/4.2* RNAi. Sadly, this antibody failed to detect a band at the expected molecular weight for NPL-4.1 in control or *npl-4.1*(RNAi) treated worms. Therefore, while we are very keen to do this experiment, the lack of an available antibody makes doing this impossible for the time being.

Minor points:

6. On page 3 line 75 there are some more comprehensive reviews of UBQLN function: PMCID: PMC7737201 and PMID: 22628307.

Thank you for pointing these out. Both have now included both references within the corresponding portion of the introduction on page 3, paragraph 2.

7. It is also worth mentioning that STI1 domains of UBQLNs help govern client binding, and that it is not simply binding of ubiquitinated protein on page 3 line 78.

This is another excellent point that we failed to mention. This has now been inserted into the corresponding text on page 3, paragraph 2.

8. There may be a missed opportunity in Figure 1 to discuss genes that have a wt lifespan phenotype which is mitigated by HSF-1 OE. Are they in the same pathway as UBQLN1 for example?

Please see our response to your major comment 3 above.

9. On page 6 line 189: the data shown have not proven an impairment of organellar protein degradation. The best way to show that would be with a pulse-chase, but that can require considerable troubleshooting and might be a major undertaking. The language just needs re-wording. This is a fair point, and we have now replaced the statement within the revised manuscript as follows:

Page 8, line 19: *“Together, our data reveal that hsf-1 OE promotes transcriptional down-regulation of genes encoding CDC-48 complex components, and a ubql-1 dependent reduction in NPL-4.1, which could lead to impairment of organellar protein degradation.”*

10. Figure 3: the x-axis label for a and c needs to be more descriptive. Is this log2 ratio?

Our apologies, these values are log2 Fold change. We have now modified all relevant axes to state this.

11. Figure 3: I would change the order to go from genes > proteins. Part e/f should go first.

We agree that this is a more logical order and have now re-arranged these panels as requested in the revised manuscript.

12. Figure 3: what are the scales for d/f? Is that a z score?

We apologise for this. No Z-score was used; the scales in the legends are log₂ Fold change. This has now been corrected in each of the corresponding Figure panels.

13. Figure 3: the graphical representation of part e is hard to understand. Why not show another volcano plot here?

We agree that the original venn was uninformative and have replaced this with a volcano plot with altered proteins labelled.

14. Figure 5 b-c don't all add up to 100? The top line of some bars is missing.

Thank you for spotting this – it has now been corrected.

15. As it stands, Figure 5d-f seems more appropriate for a Supplement since the connection is unclear and the data are mostly negative.

We have considered this, but we think that it is important to include these data within main Figure 5 as they make an important point – specifically that *hsf-1* OE animals are unable to increase lifespan in the absence of proper mitochondrial dynamics (either fusion or fission). Together with the rest of our data, this supports a working model in which ubiquilin-1-mediated changes in mitochondrial dynamics underlie the increased lifespan of *hsf-1* OE worms.

16. Supplementary Figure 1c needs to show STI1 domains of UBQL-1.

We agree and apologise for omitting this important information. The STI1 domains have now been included within our cartoon in Figure 1c.

17. In Supplementary Figure 3c there is some confusion about whether the graph is showing UBQLN-1 dependence of HSF-1 phenotype or simply KEGG pathways changed by HSF-1

We apologise for the error on our part – the title to the panel was wrong. This has now been corrected to make it clear that the panel shows pathways changed upon *hsf-1* OE in wild type animals (i.e. *hsf-1* OE v wild type (N2)). The plots comparing wildtype and *hsf-1* OE worms are now in Supplementary Figure 3a and b. The plots comparing *hsf-1* OE and *hsf-1* OE; *ubql-1(tm1574)* worms are now in Main Figure 3a and b.

18. The title of Supplementary Figure 5 is too long and not an adequate summary of the results.

We fully agree and have now modified the title of Figures 5 and S5 accordingly.

19. On Page 8 it is mentioned that it's surprising that they did not identify a greater number of proteins with altered abundance (lines 247-249). However, this could be because the proteomics done here were not with isolated cytosol as mentioned on line 247.

We take your point and have added this possible explanation to the discussion text. It is also possible that performing proteomics on whole animals could mask changes that occur in some tissues but not others. We have modified the discussion text as follows to highlight these points.

Page 11, line 17: *“It is perhaps surprising then that we did not identify a greater number of proteins exhibiting altered abundance in wildtype or hsf-1 OE worms upon loss of ubql-1. One possible explanation for this is that there may be residual UBQL-1 activity remaining in our mutant strain, as a truncated protein is produced rather than a complete knockout. Alternatively, UBQL-1 may only be critical for the turnover of a limited sub-set of proteins or the activity of other pathways, such as the autophagy lysosome system, may be increased to compensate for the chronic loss of UBQL-1 activity in our mutants. Lastly, technical aspects, such as not using cytosolic fractions, or performing proteomics on whole animals, may also have masked changes.”*

20. The discussion mentions a possible interaction between UBQLN1 and the cytoskeleton. In fact, mammalian UBQLNs were first identified as an interactor of vimentin (PMID: 10549293).

Thank you for the additional information. We were unaware of this and have now included the new point/reference within the text to reflect this.

Reviewer #2 (Remarks to the Author):

Annamary Paul Erinjeri and co-workers describes how HSF-1 promotes lifespan in *c. elegans*. Using an RNAi screens, they have identified ubiquilin-1 as a downstream factor of hsf-1. They observed that hsf-1 overexpression leads to transcriptional downregulation of all components of the CDC-48-UFD-1-NPL-4 complex, which is important to endoplasmic reticulum and mitochondria associated protein degradation. The finding that enhanced HSF1 expression increases lifespan is quite interesting but a mechanism how the HSF1 protein regulate ubiquitin expression and how ubiquitin acts (directly) on further downstream proteins remains unclear.

We thank the reviewer for their general support of the work and agree that the absence of a specific protein(s) that ubiquilin-1 acts on to influence mitochondrial morphology is a limitation of the work. However, our findings still present an important new model for *hsf-1* OE mediated lifespan extension that challenges existing paradigms within the field. While we have not defined a specific ubiquilin target that influences mitochondrial dynamics, we have expanded our analysis to give a better idea of which mitochondrial proteins exhibit altered abundance in *ubql-1* mutants (see responses below for specifics).

1. In addition, the authors switch between altered protein/gene expression levels and “activity” which is sometimes difficult to follow. One has to be careful to use the term “activity” since the authors have not tested whether genes/proteins of interest have a higher activity based on PTMs, interactions, translocation or alternative activity assays.

Thank you for pointing this out. We fully agree and acknowledge the lack of precision on our part. We have now modified the revised manuscript to more accurately reflect what is specifically changing in each experiment (i.e. *ubql-1* mutation or RNAi) throughout.

2. Why have the authors mentioned the NHR-49 as an interactor of ubiquitin? Is this factor regulated in the transcriptomics or proteomics dataset? Later the authors mentioned an affected lipid metabolism. Do the authors detect also other genes/proteins associated to fatty acid metabolism in their gene/protein expression data? It might be worse to select one affected candidate (ACS2?) and test for protein ubiquitination.

This is a very good point. We do not find any evidence that NHR-49 mRNA or protein levels are altered by *hsf-1* OE or *ubql-1* mutation in our transcriptomics/proteomics data. Furthermore, we do not see any enrichment for NHR-49 targets in differentially regulated genes. However, we do observe a decrease in the abundance of ACS-2 protein in *hsf-1* OE; *ubql-1* mutant worms. *Acs-2* expression is highly dependent on NHR-49, which lead us to assume that NHR-49 activity must be altered by loss of *ubql-1*. However, we have now performed experiments to formally test this and found that this assumption was incorrect (please see our response to your point 5 for further details).

We agree that testing the ubiquitylation status of ACS-2 would be interesting. However, as far as we are aware, there are no available antibodies against ACS-2 that would allow us to perform this targeted experiment.

In addition, as requested, we have now clarified exactly which metabolic genes and pathways are dependent on *ubql-1* in *hsf-1* OE worms. We have modified the text as follows to reflect this:

Page 7, line 5: *“The enrichment for “metabolic pathways” among hsf-1 OE mediated changes in gene expression and protein abundance prompted us to use WormFlux²⁵ to investigate whether any metabolic pathways were significantly altered by loss of ubql-1. This revealed “Fatty acid biosynthesis” (ACS-2, PPT-1; P-enrichment = 0.00096) and “nicotine and nicotinamide metabolism” (parg-2; P-enrichment = 0.038) as pathways that are disrupted in hsf-1 OE worms lacking ubql-1 (Supplementary Table 4 and Supplementary Table 7).”*

3. Have the authors tested a general change in protein ubiquitination in their worm models? either by immunoblotting or a diglycine remnant IP combined with LC-MS?

This is a great point. To address this, we have assessed levels of total polyubiquitin, K48-linked ubiquitin (proteasome targeted) and K63-linked ubiquitin (lysosome targeted) in *hsf-1* OE and *hsf-1* OE; *ubql-1* mutant worms on day 1 and day 5 of adulthood, using SDS-PAGE and western blotting. These experiments revealed that aggregated polyUb, K4-linked Ub and K63-linked Ub all increase between day 1 and day 5 of adulthood. Furthermore, while *ubql-1* mutation or *hsf-1* OE did not consistently alter levels of soluble or aggregated polyUb or K63-linked Ub proteins on day 1 or day 5 of adulthood, loss of *ubql-1* lead to an increase in K48-linked ubiquitylated proteins on day 1 and day of adulthood. Interestingly, *hsf-1* OE can suppressed K48-linked Ub protein aggregation on day 5 of adulthood, but did so even in the *ubql-1* mutant background (although loss of *ubql-1* does appear to increase absolute levels of K48-linked Ub aggregates in *hsf-1* OE worms on day 5 of adulthood). These data are consistent with our findings with polyQ in muscle tissue. These new data are included in new Fig. 2h and i and Supplementary Fig. 2f-m). We have updated the results text to reflect these new data as follows:

Page 5, line 25: “Given that polyQ is an artificial sensor of proteostasis capacity, we reasoned that loss of *ubql-1* may impede the ability of *hsf-1* OE to promote the degradation of endogenous ubiquitylated substrates with age. Therefore, we assessed the abundance of polyubiquitylated, K48-linked ubiquitylated (proteasome directed) and K63-linked ubiquitylated (lysosome directed) proteins²³, on day 1 and day 5 of adulthood.

We observed that levels of high molecular weight, aggregated polyUb, K48-ubiquitylated and K63-ubiquitylated proteins increased with age in all genotypes tested (Fig. 2h, i and Supplementary Fig. 2f-m). However, the level of soluble and aggregated polyubiquitylated or K63-ubiquitylated proteins was not consistently altered on day 1 or day 5 of adulthood in *hsf-1* OE or *ubql-1* mutant animals, compared to wild type controls (Supplementary Fig. 2f-m).

In contrast, the levels of both soluble and aggregated K48-ubiquitylated proteins was elevated in *ubql-1* mutant worms on day 1 and day 5 of adulthood (Fig. 2h, i and Supplementary Fig. h, i). Interestingly, *hsf-1* OE did not consistently alter levels of K48-Ub proteins on day 1 of adulthood, but did reduce levels of both soluble and aggregated K48-Ub proteins in both wild type and *ubql-1* mutant backgrounds on day 5 of adulthood (Fig. 2h, i and Supplementary Fig. 2h, i). These data suggest that both *ubql-1* and *hsf-1* OE suppress the accumulation and aggregation of K48-linked ubiquitylated proteins with age, but that *ubql-1* is not necessary for *hsf-1* OE to mediate these effects.”

4. In case *hsf1* has a direct effect on mitochondrial proteins, is HSF1 localized in mitochondria? Or do the authors observe changes in protein ubiquitination in mitochondria? The experiments with the protein aggregation are interesting but how correlate this finding with the remodelling of mitochondria? Is it possible to overlap the aggregate staining with the mitochondrial and or ER-network?

This is an interesting idea, however HSF-1 localisation and structure have been well-studied in *C. elegans*, and it is clear that HSF-1 is localised exclusively within the nucleus, both basally, and under stress conditions (PMID: 27688402 & PMID: 23107491).

To test whether polyglutamine aggregates are found within mitochondria, we stained polyQ worms with TMRE (to detect mitochondria) and then performed Z-stacks to determine whether aggregates were ever found within mitochondria. We found that polyQ aggregates were always localised outside of mitochondria in wildtype and *hsf-1* OE worms. We have not included these data within the paper as the finding was negative. We provide representative images below for reference. Please note that the fluorescence signal from polyQ aggregates is so strong that it bleeds into the red channel of the TMRE. However, the we were able to confidently detect the presence of aggregates as the TMRE signal did not bleed into the YFP channel.

Ultimately, we do not think that suppression of age-related aggregation and increased mitochondrial fusion are necessarily linked, as *ubql-1* knockdown suppresses one, but not the

other, in *hsf-1* OE worms. Instead, we think that these are probably independent events influencing lifespan.

NB: White arrows indicate polyQ aggregates, which we always found outside of mitochondrial networks (diffuse red signal) in both backgrounds tested.

With regards to checking for changes in ubiquitination of the mitochondrial proteome, this is also a very interesting idea. It is possible that the stability of mitochondrial outer membrane proteins, and their ubiquitylation status, is altered in *hsf-1* OE worms in a *ubq1-1* dependent manner. However, performing this experiment is not trivial, and we think that these data would be better suited to a more focused follow-up study. However, we have re-interrogated our proteomics data to specifically look for changes in the abundance of mitochondrial proteins upon *hsf-1* OE. We find that of the 417 mitochondrial proteins detected, 66 exhibit altered abundance (22 increased and 44 decreased) upon overexpression of HSF-1. 21 of these proteins are annotated as outer or inner membrane proteins. While we acknowledge that this is an indirect approach, the findings are at least consistent with a change in the stability, and possibly ubiquitylation status, of a sub-set of the mitochondrial proteome. We have included these data in new Figure S4a and S5d. We have also added the following text to the results section:

Page 8, Line 34: *“Of the 417 mitochondrial proteins identified in our proteomics dataset, 66 exhibited altered abundance in hsf-1 OE animals (FDR $P < 0.05$), with 22 increased and 44 decreased (Supplementary Fig. 4a and Supplementary Table 5). Of these, 21 proteins are annotated as outer or inner membrane proteins. These data suggest that hsf-1 OE alters the stability of a sub-set of the mitochondrial proteome, and that the increased mitochondrial fusion in hsf-1 OE worms is unlikely to be the result of a general change in mitochondrial mass.”*

5. The authors observed a reduced level of ACS2 and postulate that an altered activity of NHR49 might be responsible. However, this should be verified by some experiments. Do the author observed a general down regulation of mitochondria in *hsf-1* OE and double mutant worms? This could be tested by the levels of mtDNA and labelling all mitochondrial proteins in the volcano plot. A box plot might reveal whether these changes of mitochondrial proteins is significant (compare whole protein distribution versus all detected mitochondrial proteins in a box plot and calculate significances).

This is a very good point. Our attention was directed towards NHR-49 for two reasons: (i) recently published work demonstrated that *nhr-49* is necessary for the increased lifespan of

hsf-1 OE worms (PMID: 36261024) and increased HSF-1 activity in germline deficient worms (PMID: 37162952), and (ii) we observed that *ubql-1* is necessary for increased ACS-2 levels (a key mitochondrial membrane protein and canonical marker for NHR-49 activity) in *hsf-1* OE worms. Therefore, we assumed that our findings fed into previous observations; interestingly, this assumption appears to be wrong.

We crossed a well-established transcriptional reporter for NHR-49 activity (*acs-2* promoter driving GFP expression) into our *hsf-1* OE background, and exposed animals to *ubql-1* RNAi. As expected, *nhr-49* RNAi suppressed reporter activity in both wildtype and *hsf-1* OE backgrounds. Surprisingly, we also found that *hsf-1* OE reduced reporter activity; however, this was unaltered by the presence of *ubql-1* (RNAi). These results suggest an interesting model where loss of *ubql-1* leads to reduced ACS-2 levels independently of regulating NHR-49 activity. These new data are presented as new Supplementary Fig. 3i and j.

We have also modified the corresponding results and discussion sections as follows to reflect this:

Results, page 7, line 26: *“Acs-2 expression is strongly controlled by the transcription factor NHR-49, which has been shown to be necessary for lipid homeostasis and increased lifespan, including in hsf-1 OE worms^{9,28-33}. Therefore, we hypothesized that ubql-1 may be exerting a positive effect on lifespan by promoting NHR-49 activity.*

As expected, we found that nhr-49 (RNAi) shortens the lifespan of wildtype worms and prevents hsf-1 OE-mediated lifespan extension (Supplementary Fig. 3g and h and Supplementary Table 2). However, knockdown of ubql-1 did not alter NHR-49 activity in hsf-1 OE worms (Supplementary Fig. 3i and j), suggesting that ubql-1 does not influence ACS-2 levels or lifespan in hsf-1 OE animals through regulation of NHR-49 activity.”

Discussion, page 12, line 1: *“We also identified several proteins whose abundance decreased in hsf-1 OE worms upon loss of ubql-1. Among the most reduced proteins was the mitochondrial acetyl CoA-synthetase, ACS-2, which has a core role in fatty-acid metabolism and whose expression is controlled by the transcription factor, NHR-49^{9,53}. Given that nhr-49 and altered lipid metabolism are necessary for the increased lifespan of C. elegans possessing enhanced HSF-1 activity^{51,52}, it is interesting that we did not see any effect of ubql-1 RNAi on NHR-49 activity. However, we did find that ubql-1 was necessary for normal fat levels and elevated ACS-2 levels (a key target of NHR-49) in hsf-1 OE worms. ACS-2 plays an important role in regulating fatty-acid metabolism but is not required for normal lifespan⁵³. However, ACS-2 does mediate lifespan extension in response to lysosomal signalling in C. elegans⁵³. Future work to understand how, if at all, changes in ACS-2 and other ubql-1 dependent metabolic processes impact hsf-1 OE lifespan, will reveal whether these changes are causal to lifespan extension.”*

The idea of looking at general changes in mitochondrial protein abundance / mitochondrial mass is very nice. We have now done this (please see our response to your Point 4).

6. Since the mitochondrial remodelling is dramatically changed and the extensive remodelling of mitochondria is mentioned in the abstract, why have the authors hid this data in the supplement? Some immunostainings with a higher magnification might be helpful here to better visualize the changes of mitochondria.

We fully agree with this point and have now included representative confocal images of mitochondria in each genotype/treatment group. These are now presented as new Fig. 5c and S5c.

Minor:

1. The transcriptomics and proteomics analysis the data are not well documented in the SI material. The description of the axis of Figure 2 are not proper described. For example, one should use $-\log_{10}$ p-values and what means “difference? What is the bended line within the volcano plot? Please explain briefly at least in the method part the FDR calculation and how the bended line was generated. A more systematic analysis such as a PCA or unsupervised hierarchical clustering might be helpful to judge the significance of the observed changes.

We apologise for the lack of clarity in these figure panels and have now revised our methods section, figure panels and legends to address these points. In brief, “Difference” has been changed to “Log2 Fold change” in all volcano plots and legends. The blended lines in each volcano plot were generated automatically in Perseus to segregate all proteins with an FDR p-value < 0.05 regardless of fold change (hence the different shapes of these lines in each panel). We have now included this information in the materials and methods and appropriate legends.

2. The header/identifier of table 3 is not proper described. What means “#Name?”, difference, and what values are in column L-S? The table should contain all proteins which were quantified, independent of their ratio and p-value. This might help to judge the depth of the proteome and shows which factors are not affected. This is mandatory for the supplement.

We apologise for these oversights on our part. We have now corrected the headers/identifiers to be more informative and have included the full list of proteins detected in our mass-spec experiments as a new supplementary table. In short, “#Name?” was a cell error caused by the text “ $-\log_{10}P$ ” within the cell.

3. The colour code of protein changes in Figure 3 is not explained. What means -5 in Figure 3d?

We apologise for not including this information originally. The scales reflect Log2 fold change. This has now been added to all relevant figure panels and legends. In addition, the colour code is that red dots outside of the curved line represent proteins with significantly altered abundance (increased, righthand side; decreased, lefthand side). This is now included within the relevant figure legends.

4. Page 5 line 144-148: The authors report that loss of ubiquitin results in 43 increased and 25 decreased proteins. Then they described the RNA-Seq data as “Similarly”, which shows 10x more upregulated genes?

This is another very good point. We apologise for our loose language and have now altered the wording in the text to be more accurate.

6. The authors stated “we first investigated whether ubiquilin-1 influences HSF-1 activity. Loss of ubql-1 function did not suppress the expression of canonical HSF-1 target genes... demonstrating that HSF-1 activity is unaffected by the loss of ubql1 activity”. It is highly speculative to judge the activity by the

expression level of two target proteins. HSF1 might affect also other proteins. Have the authors identified some PTMs on HSF1 or do the authors observed an altered ubiquitination of HSF1 or other proteins?

We apologise for not being more precise with our language and for not clearly explaining why we selected *hsp-16.11* and *hsp-70* as genes to test for an indication of HSF-1 activity. Both genes are canonical HSF-1 targets that are strongly reflective of HSF-1 activity (i.e. increased expression of both genes is always associated with increased HSF-1 activity). Changes in the expression of these genes acts as a “gold standard” for measuring HSF-1 activity. However, we have now included a third HSF-1 target gene (*F44E5.4*) to our heat shock data set (new Figure 2c) to make our conclusions even more robust. As expected, the induction of all three genes is higher following heat shock in *hsf-1* OE animals. This is unaffected in *hsf-1* OE; *ubql-1* mutant worms, which strongly suggests that loss of *ubql-1* does not modify HSF-1 activity. However, we take your point that we cannot rule out more subtle or nuanced changes in HSF-1, and have therefore modified the accompanying text to reflect this as follows:

Page 4, line 35: “These data suggest that *ubql-1* mutants do not block *hsf-1* OE lifespan extension by reducing HSF-1 activity, although we cannot completely rule out more subtle or nuanced changes to HSF-1.”

Please note that we did not measure basal *F44E5.4* mRNA, as we did not detect any statistically significant difference in the basal expression of *hsp-16.11* or *hsp-70* in *hsf-1* OE or *ubql-1(tm1574)* mutant worms.

7. Do the authors detect a general change in protein ubiquitination by immunoblotting of cellular extracts or isolated organelles?

We do – please see our comments to your major point 3 for more information.

8. Do the authors observe a change in lipid metabolism in *hsf1* OE worms? It might useful to verify changes in lipid synthesis by a lipidomics/metabolomics analysis.

We fully agree that this would be very useful/informative follow-up work, but embarking on lipidomics/metabolomics lies beyond the scope of this paper. In the future, this is a direction we are keen to pursue; however, we do not think that these experiments will support or disprove the core claims made here.

Reviewer #3 (Remarks to the Author):

In this manuscript, the authors investigate how overexpression of heat shock factor, HSF-1, extends lifespan in *C. elegans*. They started with a RNAi screen and identified that *ubql-1* is required for lifespan extensions of *hsf-1* OE animals. By transcriptomics and proteomics analyses, they claimed that genes and proteins in metabolic pathway are upregulated in *hsf-1* OE animals, which is dependent on *ubql-1* and its potential action on *nhr-49*. Since UBQL-1 is involved in protein degradation they focused on proteins whose reduction in *hsf-1* OE animals was dependent on *ubql-1*, namely NPL-4.1/2. To further characterized the role of CDC-48-NPL-4-UFD in HSF-1 OE animals. Then, they turned their focus on metabolic and mitochondrial (dynamics) changes. *ubql-1* mutation in *hsf-1* OE worms restored OCR and ATP level to wild type and reduced the level of lipid droplets (measured by Oil-Red-O staining). The loss of *ubql-1* also altered muscular mitochondrial morphologies. The overexpression of HSF-1 induced an increase in the number of elongated/tubular mitochondria, while *ubql-1* mutation compromised this effect. RNAi against fission not fusion genes inhibited the lifespan extension in *hsf-1* OE worms and knockdown of mitophagy did not change the lifespan of *hsf-1* OE. While this study demonstrates a role for HSF-1 OE in mitochondrial remodeling, the role of *ubql-1* in

this process remains unclear. Further it is unclear how reduced mitochondrial fission suppress hsf-1 OE lifespan extension.

We thank the reviewer for their time and careful assessment of our work. The comments and suggestions provided have been incredibly useful and we have done our best to address all of them below.

1. Ubiquilin-1 is required for HSF-1 mediated lifespan extension- The RNAi screen is not described in enough detail and missing from the methods section. How much FUdR did the authors add? How did they evaluate the outcomes of RNAi? How did they measure the lifespan on 12-wells plates?

We apologise for this omission and have now revised our methods section to add greater details regarding the screening procedure. We have included the following text to the Methods section, Page 14, line 19:

“Screening for modifiers of lifespan in hsf-1 OE worms

RNAi screening was carried out in flat-bottomed 12-well polystyrene plates, with wells supplemented with 1 ml of standard NGM media containing 1 mM IPTG and 100 ug/ml ampicillin. Each well was seeded with 100 ul of an individual RNAi culture (prepared as described above) and allowed to dry at room temperature. Once dry, FUdR was added to each well to a final concentration of 15 uM and also allowed to dry. N2 and hsf-1 OE worms were stage-synchronised by hypochlorite treatment as described⁶⁸ and allowed to mature to the L4 larval stage on standard NGM plates. L4 worms were then washed off NGM plates and added to lifespan screening plates at a density of 20-30 per well. Worms were then scored every day for survival, with each RNAi screening plate run in triplicate for each genotype. Rare examples of progeny that escaped FUdR treatment or worms that exhibited vulval prolapse or internal hatching were removed from plates and censored. Survival was scored by gently touching individual worms with a platinum pick. Worms were scored as dead in the absence of pharyngeal pumping or a response to touching. All RNAi clones screened, and corresponding survival outcomes, can be found in Supplementary Table 1.”

1a. Since *ubq1-1* is an HSF-1 target gene, they authors should show that *ubq1-1* transcription is mediated by hsf-1. Is *ubq1-1* transcription reduced upon hsf-1 RNAi or in hsf-1 mutants?

The focus of our study is on the effects of *hsf-1* OE on *ubq1-1* expression and lifespan. The fact that HSF-1 directly binds to the *ubq1-1* promoter, and that *ubq1-1* mRNA is elevated in *hsf-1* OE worms, strongly argues for a direct action of HSF-1 on *ubq1-1* expression. We do not think that experiments in *hsf-1* RNAi or loss of function mutants would add useful information to this study. Whether *hsf-1* RNAi or loss-of-function do, or do not, reduce *ubq1-1* expression will not alter the conclusions of this study.

1b. Where is *ubq1-1* expressed, this is important since the authors are determining tissue-specific proteostasis models and mitochondrial measures.

This is a really nice point. Using the *C. elegans* tissue expression database developed by the Murphy and Troyanskaya labs (Kaletsky et al. 2018, PLoS Genet; PMID: 30096138), we find

that *ubql-1* is ubiquitously expressed in all hermaphrodite cell/tissue-types, with the exception of BAG neurons, arcade cells and valve cells. These data are included as new Figure S1e.

1c. Is *ubql-1(tm1574)* a null mutant? The authors showed this mutant has truncated mRNA. Does this mutant have truncated or no UBQL-1 protein?

This is a really good question. It is definitely not a null mutant, as we were able to detect UBQL-1 peptides (at lower levels that did not make our $p < 0.05$ FDR) in our mutants by mass-spectrometry. However, the *ubql-1* mutant does express reduced levels of a truncated mRNA, suggesting that the *ubql-1* mutant produces a truncated loss-of-function protein lacking the UBL domain as depicted in Figure S1c. We have now made this point more explicitly within the results text (Page 4, line 20) as follows:

*“In addition, we assessed lifespan in *ubql-1(tm1574)* mutants that harbour a 755bp deletion spanning the whole of exons 1 and 2, and express reduced levels of a truncated *ubql-1* mRNA and protein (Fig. 1d, Supplementary Fig. 1c and f, and Supplementary Table 3).”*

1d. Is *ubql-1* activity increased in HSF-1 OE animals? While the authors are trying to get to this with their proteomics analysis, it's in a rather confusing and roundabout way.

This is a very difficult thing to directly measure in worms, *in vivo*. Given that we observe increased *ubql-1* expression and *ubql-1* dependent changes in NPL-4.1 and ACS-2 abundance, mitochondrial morphology and lifespan in our *hsf-1* OE worms, we think that this is a logical/reasonable assumption. However, we do acknowledge that it remains possible that other molecular changes within *hsf-1* OE worms make them more dependent on *ubql-1*, without an increase in ubiquilin-1 activity.

2. Maintenance of proteostasis capacity does not require *ubql-1* function - Line 105-110: Since *ubql-1* is regulated by HSF-1, why do the authors think loss of *ubql-1* would affect HSF-1 activity? Is there any feedback effect? Does any literature support this?

We apologise for the confusion here. We do not think that *ubql-1* should regulate HSF-1 activity – there is no evidence that this should be the case in the literature. However, we wanted to formally test this and rule out the simplest possible scenario for a suppression of *hsf-1* OE lifespan upon loss of *ubql-1* before investing time in more complicated experiments.

2a. The data in Figure 2a shows that *ubql-1* mutants show an intermediate phenotype between WT and HSF-1 OE in *hsp-16.11* and *hsp-70* levels. What is the conclusion here? That the expression is equal to WT or equal to HSF-1 OE? This data needs to be strengthened.

This is a really good point. Upon re-inspection of our results, we realised that it would be more appropriate to run a two-way ANOVA on these data, as we were interested in the interaction of two different variables – *hsf-1* OE and *ubql-1* mutation. Our new two-way ANOVA reveals a significant effect of *hsf-1* OE, but not *ubql-1* mutation, on *hsp-16.11* and *hsp-70* mRNA levels post heat shock. By coupling this with pairwise comparisons using Fishers LSD test, we can now more confidently conclude that *hsp-16.11* and *hsp-70* expression is increased by *hsf-1* OE and that this is unaffected by loss of *ubql-1*. We have also added a third canonical HSF-1 target gene (*F44E5.4*) to our heat shock data, to make our conclusion even more robust. The new data and stats can be found in new Figure 2a-c. Please note that we did not measure basal *F44E5.4* mRNA, as we did not detect any statistically significant difference in the basal expression of *hsp-16.11* or *hsp-70* in *hsf-1* OE or *ubql-1(tm1574)* mutant worms.

We should also point out that there were other examples in the paper where one-way ANOVA was incorrectly used. All relevant data have now been reanalysed with two-way ANOVA

followed by Fishers LSD test. The new stats are presented in corresponding figure panels and this information has been added to all relevant figure legends.

2b. Line 115-117, “Given that *ubql-1* mutants do not exhibit increased HSF-1 activity, these data suggest that *ubql-1* is necessary for increased stress resistance in *hsf-1* OE worms” This statement is not true. Mutation of *ubql-1* in *hsf-1* OE animals did not change the survival after heat shock or the level of polyQ aggregates (fig 2c-2f). However, it did shorten the survival in wild type, suggesting that *ubql-1* is required for heat stress survival in basal condition (fig 2c).

We understand the confusion here. Our conclusion comes from the fact that *ubql-1* mutants (black dotted line) are more stress resistant than their wild type counterparts (black solid line). In fact, *ubql-1* mutants are as stress resistant as *hsf-1* OE worms (green solid line). However, *hsf-1* OE does not further increase the stress resistance of *ubql-1* mutants (green dotted line). One possible interpretation here is that loss of *ubql-1* increases HSF-1 activity; however, we did not see any evidence of this in our RT-qPCR data (Figure 2a-c and Figure S2a and b). Therefore, the alternative explanation (in genetic terms) is that *ubql-1* is necessary for *hsf-1* OE to increase stress resistance. We have changed the word “required” to “necessary” in the text to use the correct terminology here.

2c. Fig 2d-f, although the *ubql-1*(RNAi) does not affect polyQ aggregation in neither wild type nor *hsf-1* OE animals in intestine, it did affect PolyQ aggregation in body wall muscle(at least in one experiment, how many experiments were conducted in total? More repats may be necessary), did the authors try *ubql-1* mutant? Based on their lifespan analysis, the *ubql-1* mutant appears to show stronger phenotype comparing the lifespan of *ubql-1* RNAi in fig 1c-d to ctrl and that of *ubql-1* mutant with wild type in fig 4b-d.

This is a good point, and we apologise for the imprecise language used on our part. In absolute terms, it is true that *ubql-1* (RNAi) leads to a modest increase in polyQ aggregation in *hsf-1* OE muscle tissues on day 3 and day 5 of adulthood. Our point is that by day 5 of adulthood, the ability of *hsf-1* OE to suppress aggregation is unaffected compared to the corresponding wildtype *ubql-1* RNAi control group (i.e. the reduction in aggregates between WT and *hsf-1* OE worms is the same when comparing the control or *ubql-1* RNAi groups). We have now modified the text within the corresponding results section as follows to more precisely, and appropriately, describe our data:

Results, Page 5, line 14: “PolyQ aggregation increased with age in both intestinal and muscle tissues (Fig. 2e-g and Supplementary Fig. 2c-e) of wildtype worms and was strongly suppressed on day 3 and day 5 of adulthood by overexpression of *hsf-1* (Fig. 2e-g and Supplementary Fig. 2c-e). Knockdown of *ubql-1* increased polyQ aggregation on day 5 of adulthood within the muscle, but not the intestine, of wild type and *hsf-1* OE worms (Fig. 2e-g and Supplementary Fig. 2c). However, the degree to which *hsf-1* OE suppressed polyQ aggregation in muscle and intestinal tissues compared to wildtype counterparts, was the same in both the control (RNAi) and *ubql-1* (RNAi) treatment groups (Fig. 2f, g and Supplementary Fig. 2d, e). This suggests that *ubql-1* is not necessary for *hsf-1* OE to suppress age-related polyglutamine aggregation.”

All muscle and intestine polyQ experiments have now been performed at least three times, and the data presented are representative of what we observed across the three trials. However, we have not performed these experiments in the mutant background. While we

agree that *ubql-1* mutation has a slightly stronger effect on wild-type lifespan than *ubql-1* RNAi, both *ubql-1* manipulations suppressed *hsf-1* OE lifespan to the same extent. Furthermore, (as stated above) we see that *ubql-1* RNAi increases polyQ aggregation on day 5 of adulthood in wild type and *hsf-1* OE worms. Together, these findings argue that RNAi is effective and sufficient for these experiments.

3. Ubiquilin-1 promotes metabolic remodelling in *hsf-1* OE animals- Is it necessary to include the transcriptomic analysis? What is the rationale for transcriptomic analyses? Given that *ubql-1* is important for protein degradation, it seems that the proteomics should drive the manuscript.

Yes, this is a really good point, but we wanted to be as comprehensive as possible and try to understand the secondary changes that may occur downstream of UBQL-1's effects on protein stability. The RNA-seq also allowed us to determine whether *ubql-1*-dependent changes in protein abundance were likely to come from transcriptional or post-transcriptional changes. We have added text to reflect this as follows:

Page 7, line 19: *"Given that ubiquilins have a central role in protein degradation pathways associated with the cytosol/nucleus, endoplasmic reticulum (ER) and mitochondria^{13,18,26,27}, we reasoned that ubql-1 primarily promotes lifespan extension in hsf-1 OE animals by promoting the degradation of key target proteins, with transcriptional changes arising as a secondary consequence of these effects."*

3a. Why did the authors use different $-\log P$ in Fig 3a and 3c? Different criteria or statistics?

Our apologies for not making this clearer. We used the same FDR cut-off for both plots (FDR < 0.05). However, we think this confusion comes from the fitted line plotted by Perseus. We specified that any proteins with an FDR p-value < 0.05 should be segregated without considering fold change. The presence of more proteins showing small but significant changes in abundance is what changes the trajectory (and y-intercepts) of the lines on the graphs. Our specified cut-off criteria were the same in all cases.

3b. What are the Gene Ontology (GO) analyses of genes and proteins that upregulated in *hsf-1* OE but downregulated in *hsf-1* OE; *ubql-1*(-) and vice versa? By examining this, we can learn what *ubql-1* regulates in *hsf-1* OE worms.

This is a very good idea. We tried performing this analysis originally, but there are too few genes/proteins changed to detect any significant enrichment in pathways by KEGG. As an alternative, we, have performed Gene Ontology enrichment and WormFlux metabolic pathway enrichment on these genes and proteins. These new findings are included in new Supplementary Tables 4 and 7, and we have included new text to reflect our findings in the results as follows:

Page 7, line 1: *"Gene ontology analysis revealed that ubql-1 dependent genes in hsf-1 OE worms were enriched for the terms "cell projection organization" (p < 0.00018) and "defense response" (p<0.00022), while differentially regulated proteins were enriched for the term "actin-based filaments" (Supplementary Table 4 and Supplementary Table 7). The enrichment for "metabolic pathways" among hsf-1 OE mediated changes in gene expression and protein abundance prompted us to use WormFlux²⁵ to investigate whether any metabolic pathways*

were significantly altered by loss of *ubql-1*. This revealed “Fatty acid biosynthesis” (*ACS-2*, *PPT-1*; *P*-enrichment = 0.00096) and “nicotine and nicotinamide metabolism” (*parg-2*; *P*-enrichment = 0.038) as pathways that are disrupted in *hsf-1* OE worms lacking *ubql-1* (Supplementary Table 4 and Supplementary Table 7). Finally, an assessment of sub-cellular localization revealed that of the 43 proteins with altered abundance in *hsf-1* OE; *ubql-1* mutants, 13 were membrane associated (Supplementary Table 3). These results indicate that *ubql-1* regulates the expression/abundance of genes/proteins associated with diverse processes, with membrane associated and fatty-acid biosynthesis proteins affected.”

3c. What happened to a further characterization and follow up of NHR-49? The authors pose the hypothesis that *ubql-1* could alter the stability of NHR-49 complexes and spend some time in the discussion to integrate this into their data, but additional experiments linking NHR-49 activity, HSF-1 OE, *ubql-1* and the mitochondrial phenotypes are warranted.

This is a very good point. Our attention was directed towards NHR-49 for two reasons: (i) recently published work demonstrated that *nhr-49* is necessary for the increased lifespan of *hsf-1* OE worms (PMID: 36261024) and increased HSF-1 activity in germline deficient worms (PMID: 37162952), and (ii) we observed that *ubql-1* is necessary for increased ACS-2 levels (a key mitochondrial membrane protein and canonical marker for NHR-49 activity) in *hsf-1* OE worms. Therefore, we assumed that our findings fed into previous observations; interestingly, this assumption appears to be wrong.

We crossed a well-established transcriptional reporter for NHR-49 activity (*acs-2* promoter driving GFP expression) into our *hsf-1* OE background, and exposed animals to *ubql-1* RNAi. As expected, *nhr-49* RNAi suppressed reporter activity in both wildtype and *hsf-1* OE backgrounds. Surprisingly, we also found that *hsf-1* OE reduced reporter activity; however, this was unaltered by the presence of *ubql-1* (RNAi). These results suggest an interesting model where loss of *ubql-1* leads to reduced ACS-2 levels independently of regulating NHR-49 activity. These new data are presented as new Supplementary Fig. 3i and j.

We have also modified the corresponding results and discussion sections as follows to reflect this:

Results, page 7, line 26: “*Acs-2* expression is strongly controlled by the transcription factor NHR-49, which has been shown to be necessary for lipid homeostasis and increased lifespan, including in *hsf-1* OE worms^{9,28-33}. Therefore, we hypothesized that *ubql-1* may be exerting a positive effect on lifespan by promoting NHR-49 activity.

As expected, we found that *nhr-49* (RNAi) shortens the lifespan of wildtype worms and prevents *hsf-1* OE-mediated lifespan extension (Supplementary Fig. 3g and h and Supplementary Table 2). However, knockdown of *ubql-1* did not alter NHR-49 activity in *hsf-1* OE worms (Supplementary Fig. 3i and j), suggesting that *ubql-1* does not influence ACS-2 levels or lifespan in *hsf-1* OE animals through regulation of NHR-49 activity.”

Discussion, page 12, line 1: “We also identified several proteins whose abundance decreased in *hsf-1* OE worms upon loss of *ubql-1*. Among the most reduced proteins was the

mitochondrial acetyl CoA-synthetase, ACS-2, which has a core role in fatty-acid metabolism and whose expression is controlled by the transcription factor, NHR-49^{9,53}. Given that nhr-49 and altered lipid metabolism are necessary for the increased lifespan of C. elegans possessing enhanced HSF-1 activity^{51,52}, it is interesting that we did not see any effect of ubql-1 RNAi on NHR-49 activity. However, we did find that ubql-1 was necessary for normal fat levels and elevated ACS-2 levels (a key target of NHR-49) in hsf-1 OE worms. ACS-2 plays an important role in regulating fatty-acid metabolism but is not required for normal lifespan⁵³. However, ACS-2 does mediate lifespan extension in response to lysosomal signalling in C. elegans⁵³. Future work to understand how, if at all, changes in ACS-2 and other ubql-1 dependent metabolic processes impact hsf-1 OE lifespan, will reveal whether these changes are causal to lifespan extension.”

4. Ubiquilin-1 promotes down-tuning of endoplasmic reticulum and mitochondrial associated degradation components in hsf-1 OE animals

4a. Line 173: “Therefore, we focused our attention on proteins whose levels were elevated in hsf-1 OE animals upon reduction of UBQL-1 function. Among the proteins whose levels are decreased upon hsf-1 OE, we identified 4 whose reduction was dependent on UBQL-1.” This description is confusing. The authors should consider rephrasing: to something like: Since we hypothesize that UBQL-1 activity is increased in HSF-1 OE animals, which would lead to increased degradation of UBQL-1 target proteins, we focused our attention on proteins whose levels were decreased in hsf-1 OE animals in a manner dependent on UBQL-1 function.”

We are grateful for the suggestion and apologise for the poor wording on our part. We have now revised the wording within the text as follows:

Page 8, line 2: “As we hypothesized that UBQL-1 activity is increased upon hsf-1 overexpression, we predicted that this would promote the degradation of UBQL-1 targets in hsf-1 OE worms. We therefore reasoned that proteins whose levels were increased in hsf-1 OE animals upon loss of ubql-1, may be UBQL-1 clients.”

4b. The authors conclusion that HSF-1 OE worms are long-lived due to “mild impairment of organellar protein degradation” is not convincing. Can the authors show that “mild impairment of organellar protein degradation” can extend lifespan?

This is a great point, and we agree that this would be a nice thing to show. We started performing this work using RNAi dilutions (neat, 2-, 5-, 10-, 20-, 50-, and 100-fold in L4440 (empty vector) bacteria) to “tune” the knockdown *cdc-48.1* (RNAi), either throughout life, or exclusively in adulthood. We began by focusing on *cdc-48.1* as this is the core component of ERAD and MAD and produced the most severe lifespan shortening effects in our previous trials (Figure 4h). We decided to expose animals to *cdc-48.1* RNAi dilutions throughout life, or exclusively in adulthood, as we reasoned that it is possible that (i) *cdc-48.1* expression may not be reduced in *hsf-1* OE animals before adulthood and (ii) the stage at which *cdc-48.1* is knocked down could have a bearing on lifespan outcomes.

Our preliminary data (two independent trials; n > 70 per group) has revealed that even a 100-fold dilution of *cdc-48.1* RNAi shortens lifespan in wildtype worms when administered

throughout life. However, in contrast, the same dilution of *cdc-48.1* RNAi modestly increases lifespan (10-15%) when administered specifically from the middle of the last larval stage (L4) onwards (see representative graph below).

This is an exciting proof-of-principle; however, we realised that this series of experiments will take a long time to complete and will produce a very large body of data that lends itself better to a follow-up study. For example, we also want to perform RNAi in a tissue-specific manner and repeat all of these experiments using *ufd-1* and *npl-4.1* RNAi. Lastly, it would also be necessary to perform these experiments in different contexts, such as *hsf-1* OE, *ubql-1* OE, etc, to see how this influences lifespan phenotypes upon mild disruption of *cdc-48.1/ufd-1/npl-4.1*.

Post-developmental Treatment	EV	Cdc48.1 NEAT	1:1	1:5	1:10	1:20	1:50	1:100
N =	86	93	94	94	75	92	71	84
Median survival (days)	15	11	11	13	13	15	15	17

Given that addressing this comment would go beyond the scope of this manuscript, we propose to modify our language throughout the text and refer to existing work as evidence that this may be possible. Specifically, it has previously been shown that *cdc-48.1* mutants promote lifespan extension when combined with a mutation in the deubiquitylase, *atx-3* (PMID: 21317884). As such, there is a precedent that inhibition of a core component of organellar protein degradation can increase lifespan under certain contexts. We now include this point within the discussion section as follows, and argue that future work to address this in the context of *ubql-1/hsf-1* OE, will be important:

Page 11, Line 29: “Our findings support a model in which the mild down-tuning of this complex leads to cellular adaptations that are beneficial for lifespan. Interestingly, *cdc-48.1* mutants have been shown to increase *C. elegans* lifespan by 50% when accompanied by loss of the deubiquitylase, *atx-3/ATXN3*⁵⁰, suggesting that down-tuning of CDC-48 activity can be beneficial. Future work to understand exactly when, where and by how much CDC-48.1 activity needs to be reduced to extend lifespan, will reveal the contexts in which inhibition of CDC-48 and its co-factors might be harnessed for beneficial effects.”

In addition, we have performed experiments to see whether *npl-4.1* RNAi is sufficient to increase mitochondrial fusion in wildtype worms. We find that knocking down *npl-4.1* increases

mitochondrial fusion in wildtype animals to a similar extent as *hsf-1* OE. However, *npl-4.1* RNAi does not further increase fusion in *hsf-1* OE worms. These data are consistent with our proposed model, whereby reduced levels of NPL-4.1 alter mitochondrial network dynamics upon *hsf-1* OE. We present these data in new Figure 5e.

4c. The authors need to explain the function of the CDC-48/NPL-4/UFD complex better in both the context of ERAD and MAD (and show that this function is changed upon NPL-4 reduction with and without HSF-1 OE)

We have now expanded the text within the results section as follows to better explain the role of CDC-48-NPL-4-UFD in ERAD and MAD:

Page 8, line 6: “NPL-4.1 is a central component of the CDC-48-NPL-4-UFD1 complex, which is at the core of both ER associated protein degradation (ERAD) and mitochondria associated protein degradation (MAD)³⁴. In both contexts, the CDC-48 AAA-ATPase complex acts to “pull” ubiquitylated substrates free from organelle membranes³⁵. In ERAD, this process is facilitated by NPL4 and UFD1, which bind to one side of the CDC-48 hexamer and interact with ubiquitin chains, thereby directing polyubiquitinated proteins to the CDC-48 complex for extraction³⁵. While less well-studied, NPL4-UFD1 are also necessary for the role of CDC-48 in MAD, presumably through similar functions as in ERAD^{36,37}

Unlike NPL-4.1, we did not observe an increase in the abundance of any other CDC-48 complex components in *hsf-1* OE; *ubql-1* mutants (Fig. 3f). However, we did observe a strong reduction in *npl-4.1*, *npl-4.2*, *ufd-1*, *cdc-48.1* and *cdc-48.2* mRNA levels in *hsf-1* OE worms compared to wild type animals, although transcript levels were not restored in *ubql-1* mutants (Fig. 3e). Together, our data reveal that *hsf-1* OE promotes transcriptional down-regulation of genes encoding CDC-48 complex components, and a *ubql-1* dependent reduction in NPL-4.1, which could lead to impairment of organellar protein degradation.”

We also fully agree that we have not formally demonstrated that MAD is impaired in *hsf-1* OE animals +/- *npl-4.1/4.2* RNAi. We are very keen to do these experiments, but they are deceptively difficult. To our knowledge, no MAD reporter currently exists within *C. elegans*, with the majority of MAD clients identified in yeast. In addition, antibodies or tagged lines for known MAD substrates in yeast (FZO-1 and TOMM-70) were not available. As a consequence, we were unable to directly monitor MAD in *hsf-1* OE worms.

To try to shed some light on this, we re-interrogated our proteomics data to specifically look for changes in the abundance of mitochondrial proteins upon *hsf-1* OE. We find that of the 417 mitochondrial proteins detected, 66 exhibit altered abundance (22 increased and 44 decreased) upon overexpression of *hsf-1*. 21 of these proteins are annotated as outer or inner membrane proteins. However, loss of *ubql-1* only reversed this change for one mitochondrial protein, ACS-2. While we acknowledge that this is an indirect approach, these observations are at least consistent with a change in mitochondrial associated degradation pathways in *hsf-1* OE worms. We have included these data in new Figure S4a and S5d. We have also added the following text to the results section:

Page 8, Line 34: *“Of the 417 mitochondrial proteins identified in our proteomics dataset, 66 exhibited altered abundance in hsf-1 OE animals (FDR $P < 0.05$), with 22 increased and 44 decreased (Supplementary Fig. 4a and Supplementary Table 5). Of these, 21 proteins are annotated as outer or inner membrane proteins. These data suggest that hsf-1 OE alters the stability of a sub-set of the mitochondrial proteome, and that the increased mitochondrial fusion in hsf-1 OE worms is unlikely to be the result of a general change in mitochondrial mass.”*

We also softened our language throughout the manuscript to reflect the fact that we have not directly shown a *ubql-1/npl-4.1*-dependent impairment in MAD in *hsf-1* OE worms.

4d. The authors are hypothesizing that HSF-1 OE animals have increased UBQL-1 function which leads to increased degradation of NPL-4., which means that HSF-1OE, *ubql-1* mutants should have restored longevity after depletion of NPL-4. The authors however show a further decrease in lifespan. While it is possible that this reviewer is missing something here, in any case this section needs to be much better explained.

We can fully appreciate that this result is not very logical at first glance and agree that this result should have been more thoroughly/clearly discussed/explained in the manuscript.

There are two possible outcomes for the lifespan experiments involving *hsf-1* OE; *ubql-1* mutants and *npl-4.1* RNAi. The simplest/most logical prediction is that the presence of the *ubql-1* mutation would suppress the sensitivity of *hsf-1* OE worms to *npl-4.1* (RNAi) by increasing NPL-4.1 protein levels (as you correctly point out). However, if we consider that loss of *ubql-1* is unlikely to stabilise NPL-4.1 protein levels indefinitely in the face of prolonged *npl-4.1* RNAi, then a second likely possibility arises, specifically, that without a fully functional *ubql-1* present, *hsf-1* OE worms would become further sensitised to the toxicity caused by *npl-4.1*(RNAi). Furthermore, this effect would not be as pronounced in *ubql-1* mutants without *hsf-1* OE, as these animals would have intact *cdc-48.1* and *ufd-1* expression to compensate for the chronic knockdown of *npl-4.1*. We have modified the results text concerning this experiment as follows, in an attempt to make this point more clearly/explicitly:

Page 9, line 14: *“In addition, loss of ubql-1 further shortened the lifespans of hsf-1 OE mutants exposed to npl-4.1(RNAi). This likely reflects the fact that loss of ubql-1 is unlikely to indefinitely maintain/increase NPL-4.1 protein levels in hsf-1 OE worms in the face of prolonged RNAi against npl-4.1. As such, the absence of a fully functional ubql-1, combined with reduced expression of cdc-48.1 and ufd-1, leaves hsf-1 OE animals more vulnerable to npl-4.1 knockdown. Together, these data suggest that HSF-1 overexpression reduces CDC-48-NPL-4-UFD-1 function, and that this is accompanied by a more fused mitochondrial network and altered metabolic homeostasis.”*

5. HSF-1 overexpression promotes lifespan by altering mitochondrial dynamics
- The quality of mitochondrial images in sup Fig 4c and Fig 5d should be improved, since the authors largely determined whether *ubql-1* mutation induces mitochondrial fragmentation in *hsf-1* OE animals or not based on the images. Using confocal microscopy should get a higher resolution image.

We fully agree and have now included confocal microscopy images that are representative of the data provided. These can be found as new Figure. 5c and S5c.

5a. How many repeats of mitochondrial morphology experiments were done? No error bars were shown in quantification results and the authors didn't mention how many repeats in either legends nor methods.

We apologise for not including this information. Each of these experiments was performed at least three times. We have now plotted the mean levels of mitochondrial fusion across our replicates (with error bars and statistics). This data can be found in new Figure 4d, 5b and e, and S5b.

5b. Why did they focus on the region between the pharynx and vulva or the vulva and tail for muscular mitochondria quantification? Because of these regions show vigorous mitochondrial activity? Or is it just easier to image comparing with other regions? They use body wall muscle promoter (*myo-3*) to drive mitochondrial GFP and should observe GFP all over the whole body in muscle.

You are absolutely correct that the mitGFP is expressed in all body wall muscle tissues. We focused our scoring on the region specified as it is easier to image and less sensitive to fragmentation during mounting (this approach is standard practice in *C. elegans*). We have now highlighted this point within the methods section as follows:

Page 17, line 1: *"While GFPmit is expressed in all body wall muscle cells, the region between the pharynx and vulva or the vulva and tail were selected for viewing muscle mitochondrial networks as these areas are less sensitive to mitochondrial fragmentation during mounting."*

6. Rigor:

6a. Some of the experiments (e.g. fig 2e-f, sup fig 2c-e, sup fig 4e, and sup fig 5b & c) were only done by two repeats. We strongly recommend the authors to add at least one more repeat.

This is a very reasonable request, and we have now performed all of the experiments for the panels listed at least three times. The data included in the paper are representative of the three trials run.

6b. The quantification results (bar charts) in fig 5b and 5c are exactly the same in sup fig 5e and 5f?

We apologise profusely for this oversight. We had accidentally duplicated the graphs. The correct graphs are now in new Figures 5a and S5a. The duplicated graphs (which were supposed to show repeat trials) have now been removed entirely and replaced with graphs showing the mean values across all experimental replicates. These are presented in new Figure 5b and S5b.

7. Minor comments

7a. What is the lifespan after knocking down mitochondrial fission/fusion genes in *hsf-1* OE; *ubql-1(tm1574)*?

We did not perform this epistasis experiment as we thought that our molecular data were more useful for drawing conclusions. We think that our combined findings that (i) loss of *ubql-1* suppresses *hsf-1* OE lifespan, (ii) loss of *ubql-1* suppresses mitochondrial fusion in *hsf-1* OE worms, and (iii) *hsf-1* OE does not increase lifespan in the presence of *drp-1* and *fzo-1* RNAi,

suggest that mitochondrial dynamics are an important facilitator of *hsf-1* OE-mediated lifespan extension.

7b. Does UBQL-1 affect mitochondrial dynamics by regulating fission/fusion genes?

We had the same idea, but sadly, we did not observe any sign of this within our RNA-seq or proteomics data sets and therefore do not think this is the case.

7c. Fig 2d, we encourage the authors to add *ubql-1*(RNAi) in figure to help reader better understand it is RNAi not mutant even the authors mentioned in legends.

Thank you for the helpful suggestion. We have now added this to the relevant figure panels.

7d. Fig 2 e and 2f, the authors can put full text of intestine and body wall muscle in figures, not just using initial i or m to present it.

Thank you for the good suggestion. We have now added this to the relevant figure panels.

7e. Fig 3, we encourage that the authors put the names of those proteins that are highly differentiated in volcano plots.

Thank you for the helpful suggestion. We have now added labels to our volcano plots.

Reviewer #4 (Remarks to the Author):

We thank the reviewer very much for taking the time to provide useful and constructive feedback on our work. We hope they are enjoying taking part in, and learning about, the peer-review process.

Point-by-point responses to reviewers comments

Reviewer #1 (Remarks to the Author):

The authors of "HSF-1 promotes longevity through ubiquitin-1 dependent mitochondrial network remodeling" have gone through extensive revision of their manuscript with considerable editorial changes and experimental additions. The manuscript only needs minor editorial revision.

Regarding the original major points, all of my concerns have been addressed. In particular, the addition of polyubiquitin blots enriches the study by adding an understanding of how different forms of protein degradation (the proteasome with K48 blots and autophagy with K63 blots) are influenced by hsf-1 and ubql-1.

We thank the reviewer for taking the time to consider our revised work and are happy that our efforts to improve the manuscript were appreciated. We are grateful for the reviewer's helpful comments throughout and pleased that they have recommended our work for publication.

Regarding minor point 10, Figure 3 could use minor additional clarification on which direction the log₂ ratios are measuring in (a) and (b), for ease of interpretation. If the log₂ ratio is hsf-1 OE;ubqln1(tm1574) / hsf-1 OE then it would be helpful to include that, as it clarifies that genes/proteins on the positive side are accumulated upon ubqln1 perturbation.

As suggested by the reviewer, we have now revised Figure panels 3a and 3b to state that the log₂ ratio is obtained from a comparison of the hsf-1 OE;ubqln1(tm1574) / hsf-1 OE strain.

On page 8 Figure 3f and 3e are discussed/introduced in non-chronological order.

This has now been corrected on page 8 (lines 12-18) to read:

"We observed a strong reduction in npl-4.1, npl-4.2, ufd-1, cdc-48.1 and cdc-48.2 mRNA levels in hsf-1 OE worms compared to wild type animals, although transcript levels were not restored in ubql-1 mutants (Fig. 3e). Unlike NPL-4.1, we did not observe an increase in the abundance of any other CDC-48 complex components in hsf-1 OE; ubql-1 mutants (Fig. 3f). Together, our data reveal that hsf-1 OE promotes transcriptional down-regulation of genes encoding CDC-48 complex components, and a ubql-1 dependent reduction in NPL-4.1, which could lead to impairment of organellar protein degradation."

Reviewer #2 (Remarks to the Author):

Although the authors have still not found a direct mechanism, all of the reviewer's questions have been sufficiently answered.

We are grateful to the reviewer for recognising the importance of our core findings and agree that a direct mechanism has eluded us. However, we hope that this body of

work will act as a catalyst for ourselves and others to address this in the future. We thank the reviewer for recommending our work for publication.

Reviewer #3 (Remarks to the Author):

The authors have made a commendable effort in addressing the reviewers' comments, resulting in a substantially improved manuscript. Overstatements have been appropriately removed. While they added some new experiments, they largely sidestepped the more extensive experimental work suggested by the reviewers, justifying this approach in their responses. Nonetheless, the revisions have strengthened the manuscript overall, and I recommend it for publication.

We are very grateful to the reviewer for taking the time to provide useful and thoughtful feedback throughout, and for recommending our work for publication.

Reviewer #4 (Remarks to the Author):

We are very grateful to the reviewer for taking the time to provide helpful feedback on our manuscript. We hope it has been a useful exercise for them and wish them well in their future research career.